# Antimicrobial Resistance and Clonal Lineages of *Escherichia coli* from Food-Producing Animals

**DOI:** 10.3390/antibiotics12061061

**Published:** 2023-06-15

**Authors:** Adriana Silva, Vanessa Silva, José Eduardo Pereira, Luís Maltez, Gilberto Igrejas, Patrícia Valentão, Virgílio Falco, Patrícia Poeta

**Affiliations:** 1Microbiology and Antibiotic Resistance Team (MicroART), Department of Veterinary Sciences, University of Trás-os-Montes and Alto Douro (UTAD), 5000-801 Vila Real, Portugalvanessasilva@utad.pt (V.S.); 2Department of Genetics and Biotechnology, University of Trás-os-Montes and Alto Douro (UTAD), 5000-801 Vila Real, Portugal; 3Functional Genomics and Proteomics Unit, University of Trás-os-Montes and Alto Douro (UTAD), 5000-801 Vila Real, Portugal; 4Associated Laboratory for Green Chemistry (LAQV-REQUIMTE), University NOVA of Lisboa, 2829-516 Lisbon, Portugal; 5Veterinary and Animal Research Centre (CECAV), University of Trás-os-Montes and Alto Douro (UTAD), 5000-801 Vila Real, Portugal; jeduardo@utad.pt (J.E.P.);; 6Associate Laboratory for Animal and Veterinary Sciences (AL4AnimalS), 5000-801 Vila Real, Portugal; 7REQUIMTE/LAQV, Laboratório de Farmacognosia, Departamento de Química, Faculdade de Farmácia, Universidade do Porto, R. Jorge Viterbo Ferreira, No. 228, 4050-313 Porto, Portugal; 8Chemistry Research Centre (CQ-VR), University of Trás-os-Montes and Alto Douro (UTAD), 5000-801 Vila Real, Portugal

**Keywords:** *Escherichia coli*, ESBLs, antimicrobial resistance, livestock animals, clonal lineages

## Abstract

*Escherichia coli* are one of the most important pathogenic bacteria readily found in the livestock and widely studied as an indicator that carries drug-resistant genes between humans, animals, and the environment. The use of antimicrobials in the food chain, particularly in food-producing animals, is recognized as a significant contributor to the development and spread of antimicrobial resistance (AMR) and resistance genes can be transferred from the farm through the food-chain. The objective of this review is to highlight the background of the antimicrobials use in food-producing animals, more specifically, to study clonal lineages and the resistance profiles observed in *E. coli*, as well as in extended spectrum beta-lactamases (ESBL) producing *E. coli,* in a set of food-production animals with greater relevance in food consumption, such as pigs, poultry, cattle, fish farming and rabbits. Regarding the prevalence of ESBL-producing *E. coli* among farm animals, high-to-moderate prevalence was observed, and the highest resistance rates to tetracycline and ampicillin was detected in different farms in all geographic regions. Worldwide pandemic clones and high-risk zoonotic *E. coli* clones have been identified in most food-producing animals, and some of these clones are already disseminated in different niches, such as the environment and humans. A better understanding of the epidemiology of *E. coli* and ESBL-producing *E. coli* in livestock is urgently needed. Animal production is one of the major causes of the antibiotic resistance problem worldwide and a One Health approach is needed.

## 1. Antibiotics Use in Food-Producing Animals

Since the 1950s, antibiotics have been routinely used in farm animal production during intensive farming to keep animals healthy and to increase farm productivity [1]. However, the use of antibiotics exerts selective pressure on bacterial populations, leading to antibiotic resistance [1,2]. Antimicrobial resistance is a major threat to human and animal health, and since antibiotics are widely used in food-producing animals, antimicrobial resistance has emerged globally with consequent concerns for both veterinary and human medicine [3,4]. The intensive use of antibiotics in food-producing animals can contribute to the development and spread of antibiotic resistant bacteria, which can be transferred to human populations through food products, animals, or the environment (Figure 1). When infections caused by these bacteria occurs, it can be more challenging and difficult to treat if antibiotics are used in both animals and in human medicine [4,5]. In 2017, the World Health Organization’s (WHO) Global Action Plan on Antimicrobial Resistance (GAP), in collaboration with the Food and Agriculture Organization and the World Organization for Animal Health, adopted a plan to control antimicrobial resistance through various interventions and a “One Health” approach, aiming to reduce unnecessary antibiotic use in food-producing animals, and restrict the consumption of medically important antibiotics [6,7].

With the increase in the human population and changes in consumer trends, there has been a substantial increase in global meat production [8]. It is projected that meat production will increase by 76% by 2050 with developing countries expected to see an increase from 29–35% by 2030 and 37% by 2050 [9,10]. The use of antibiotics has also increased, and global antibiotic consumption could increase by 110,000 tons by 2030 [2,10]. Antibiotic consumption in animals is not limited to therapeutic use; they are used for metaphylaxis, when the administration of an antibiotic involves the group of animals, and when some group animals are diagnosed with a disease or show clinical symptoms, with the goal being to prevent the spread of the disease. Whereas, for prophylaxis, antibiotics are used when the use involves the mass administration of an antibiotic to healthy animals, and when the risk of developing a specific disease is established. This measure is taken to prevent diseases from occurring. In the use of antibiotics as growth promoters, subtherapeutic doses of antibiotics were administrated to animals to boost feed efficiency and increase weight gain [9,10]. In the 1960s, the Swann Commission recognized a connection between the occurrence and emergence of multi-drug resistance and the use of antibiotics as growth promoters. In 1999, the European Union banned the use of four classes of antibiotics as growth promoters, followed by a complete ban on all growth promotion classes of antibiotics in food-producing animals in 2006, aiming to prevent resistance development and remove antibiotic residues from meat [11,12]. New regulations, starting in 2022, include a ban on the preventive use of antibiotics in groups of animals, restrictions on metaphylaxis treatment antibiotics, a reinforced ban on growth promotion, obligations of member states to collect data on the sale and use of antibiotics, reservation of certain antibiotics for human use only, and a ban on imported meat raised using growth promotors [7,13].

The animal production industry can play a crucial role in the emergence and transmission of antimicrobial resistance, and the overuse and misuse of antibiotics in livestock feed are considered major causes of this problem, leading to a rapid increase in antimicrobial-resistant bacteria and antibiotic residues in food and the environments [10,14]. A significant percentage of 50–80% of antibiotic use is attributed to livestock, particularly in poultry, swine and in dairy cattle, with high-resistance rates to tetracyclines, sulfonamides, and penicillins [14]. Many of these antibiotics used in human health are also used in veterinary medicine and the livestock industries. In 2018, tetracyclines accounted for 66% of antibiotic sales for livestock, followed by penicillins (12%), macrolides (8%), sulfonamides (5%), aminoglycosides (5%), lincosamides (2%), cephalosporins (1%), and fluoroquinolones (<1%) [7]. Generally the antibiotics used therapeutically are administrated orally or by injection, but data from the US indicates that significant quantities are administrated in animal feed [15,16].

Swine are considered one of the most important farm animals in terms of number and biomass, with the largest swine producers located in China, the USA, Germany, Spain and Vietnam [10]. The United Nation’s Food and Agriculture Organization predicts a projected increase in pig production of up to 8.6% by 2030 and 12.7% by 2050, as it represents a crucial source in the food chain supply with a high economic impact [10,17]. Commonly used antibiotics in pigs include amoxicillin, fluoroquinolones, penicillin, tetracyclines, cephalosporins, lincosamides, tulathromycin, polymyxin (colistin) and macrolides (tylosin), but the most frequently used classes for therapeutic purposes are penicillins and tetracyclines [1,10]. Studies conducted on pigs farms have shown an abundance of antibiotic-resistant genes in the pig microbiome, and the level of resistance to tetracycline is naturally elevated even in animals raised without the presence of tetracycline [17,18]. In poultry, the most commonly used antibiotics for disease prevention and growth promotion are virginiamycin, bacitracin, salinomycin, and tilmicosin [10,19]. A high prevalence of ciprofloxacin-resistant bacteria and antibiotic-resistant genes to chloramphenicol, quinolones, tetracyclines, and sulfonamides have been detected in poultry [10]. In cattle, a variety of antibiotics are used, such as aminoglycosides, β-lactams, chloramphenicol, fluoroquinolones, glycolipid, ionophores, macrolides, quinolones, streptogramins, sulfonamides, and tetracyclines [10]. This review focuses on the investigation of *E. coli* isolates among a variety of food-producing animals, which are considered potentially risky for the dissemination of ESBL-producing *E. coli* and, therefore, could pose significant issues for human and animal health [20,21]. 

## 2. Food-Producing Animals as One of the Potential Sources of Food-Borne Pathogens: *Escherichia coli*

Food-producing animals are the major reservoirs for many foodborne pathogens with the ability to cause diseases, sporadic illness, chronic complications, and death, such as *Campylobacter*, *Salmonella enterica*, *Escherichia coli* and *Listeria monocytogenes* [8]. The emergence of antibiotic resistance in food-producing animals leads to public health issues in clinical settings, animal husbandry, and food industry. The options for treating infections are increasingly scarce due to an exponential increase in resistant and also multi-resistant strains [7,10,22].

*E. coli* is a Gram-negative bacterium and the most prevalent facultative anaerobic species in the gastrointestinal tract of humans and warm-blooded animals. While it is normally considered harmless, some strains can be highly pathogenic [23,24] and acquire pathogenic or virulence factors, making them medically important. They can cause a number of significant illnesses and severe infections both in humans and animals, such as bacteremia, wound infections, urinary tract infection, and gastrointestinal tract infections [9,24,25]. This virulent and resistant bacteria can be transmitted from farm animals to humans through numerous pathways, including direct contact, contact with animal excretions, or via the food chain [7,25]. The increase in *E. coli* occurrence has been reported in diverse ecological niches, and its widespread availability allows for comparisons across relevant populations. It is considered a useful alert system and serves as a model for studying the emerging resistance in livestock and its possible spread to animal-derived food [22,26]. Both commensal and pathogenic *E. coli* have also been implicated in transferring resistance genes to other bacteria and can act as reservoirs of antibiotic resistance genes that may be transferred between bacterial species, including pathogenic ones. *E. coli* can act as a donor and recipient of resistance genes, thereby acquiring and passing on resistance genes to other bacteria [9,25,26]. Many resistance genes have been identified in *E. coli* isolates in recent decades, and a significant number of these resistance genes were acquired through horizontal-gene transfer [25].

Phylogenetic analyses have shown that *E. coli* strains belong to several distinct phylogenetic groups and can be classified into four main groups (A, B1, B2, and D) [27,28,29]. According to a classification by Lecointre et al. [30], groups A and B1 are considered sister groups, usually comprising commensal strains that are able to persist in the environment. On the other hand, group B2 is included in an ancestral branch and includes strains that can cause virulent extraintestinal infections. These virulent strains also belong, to a lesser extent, to phylogenetic group D [27,29]. Phylogroups B2 and D contained more virulence factors than strains belonging to phylogroups A and B1 [29]. Phylogroups differ in their ecological niches, life history, and in characteristics, such as the ability to exploit different sugar sources, antibiotic resistance profiles, and growth rates [29]. Understanding the phylogenetic group of *E. coli* provides a better understanding of how pathogenic strains acquire virulence genes [28].

An increase in the prevalence of ESBLs compromises treatment effectiveness and increases morbidity and mortality [31]. ESBLs are plasmid-mediated enzymes that hydrolyze the broad-spectrum β-lactam ring and confer antimicrobial resistance to β-lactams (penicillins, first-, second- and third-generation cephalosporins and aztreonam) but not cephamycins and carbapenems [24,32,33]. Beta-lactamases are divided into four different classes (A to D) based on their amino acid sequences; according to the Ambler system: class A (TEM, SHV and CTX-M β-lactamases), class C (CMY, DHA, and ACT) and class D (OXA) have a water molecule to hydrolyze β-lactam antibiotics, while class B utilize zinc ions to attack the reactive β-lactam ring and is characterized as metallo-β-lactamase [22,33]. The most common genetic type is CTX-M family of ESBLs, a heterogenous and complex group of enzymes that possibly derive from the relocation of chromosomal Kluyvera genes to mobile genetic elements, which is an important cause of transferable mult-drug resistance in *E. coli*. They confer the resistance of third-generation cephalosporins, such as cefotaxime and ceftazidime [22,32]. It has been reported that extended-spectrum-β-lactamase and metallo-β-lactamase-producing bacteria are common in animals and present in their environment. In recent years, the increase in animal infections due to extended-spectrum cephalosporin resistance has become worrisome [22,24]. Antibiotic resistance of *E. coli* was found to be nearly 70% against streptomycin sulfsoxazole-tetracycline. There was also a decline in the susceptibility to other antibiotics, such as ampicillin, kanamycin, sulfsoxazole, streptomycin, tetracycline, and ticarcillin [24].

In food-producing animals, the presence of ESBL-producing *E. coli* has been widespread, and there are several studies that have shown their presence in livestock [22,34]. (Table 1). A study conducted by Gaëlle Gruel et al. in Guadeloupe focused on the prevalence of ESBL-producing *E. coli* in pigs, beef cattle and poultry. The study demonstrated a moderate prevalence of ESBL-producing *E. coli* in pig and beef cattle production systems, suggesting that the implementation of an antimicrobial resistance plan for the use of antibiotics in veterinary medicine in the region may have a positive impact on the rational use of antibiotics. In contrast to pigs and beef cattle, the study observed a high frequency of ESBL-producing *E. coli* in broiler chickens. Despite the rational use of antibiotics in Guadeloupe island, *E. coli* that were resistant to third-generation cephalosporins were also found [35]. The prevalence observed in the study conducted in Guadeloupe is closer to that observed in Italy, where 29.4% of fecal and cecal intestinal content samples from cattle and 27% of samples from pigs, and 43.6% of samples from poultry were positive for ESBL-producing *E. coli.* The prevalence of ESBLs in poultry was also quite high, while in pigs and cattle it was considered moderate [36]. In a study conducted by Linda Falgenhauer et al. in Ghana, only poultry fecal samples were studied, and out of 140 broilers, 41 (29%) were considered positive for ESBL-producing *E. coli* [37]. A high prevalence continues to be observed in samples from broilers and pigs. In a study carried out by Sien De Koster et al. in Belgium, the presence of ESBL was verified in 58.4% of samples from broilers and 48% of samples from pigs. India poultry samples continue to have a higher presence of ESBL-producing bacteria than the pig and cattle samples [38]. In Nigeria, the same pattern was observed, and the presence of ESBL-producing *E. coli* was observed in 10.8% of broiler samples and 8% in spent layers samples [39]. In South Korea, samples of chickens, pigs and cattle were also studied, and the same pattern of high prevalence of ESBLs in chickens was observed. The prevalence of ESBL in food animals in this study was found to be higher than those reported in previous studies, with 94.1% in chickens, 69.5% in pigs, and 7.0% in cattle [40]. In another study carried out in Hungary by Bence Balázs et al., different results were found compared to those observed so far. The prevalence of ESBL-producing isolates was markedly different between porcine and poultry samples, with 72 (72.0%) ESBL-producing isolates found in porcine samples and 39 (34.2%) in poultry samples [41]. In a study conducted in Reunion Island, a collection of samples from cattle, pig, poultry, rabbit, and small ruminant (sheep and goat) farms on the island were obtained. Among 88 ESBL-producing *E. coli* isolates, 50 were obtained from poultry, 33 from pigs, only 2 from cattle and 3 from small ruminants, such as sheep and goats. No ESBL-producing *E. coli* were detected in rabbits. These findings highlight the presence of ESBL-producing *E. coli* in various livestock species, with poultry and pigs showing high prevalence rates, 70% in poultry farms and 50% in pig farms [42]. In another study, rabbits (51/100; 51%), swine (51/100; 51%) and poultry (110/200; 55%) were analyzed. Out of the 212 isolates of *E. coli* (53%) tested, no ESBL-producing *E. coli* strains were detected in rabbits and swine. The prevalence of ESBL-producing *E. coli* in poultry was 5.5%. These findings suggest that while ESBL-producing *E. coli* strains were present in poultry, they were not detected in rabbits or swine in this study [43]. The increasing detection of several ESBL-producing *E. coli* in fish farming has been reported to demonstrate resistance against those antibiotics in water and fish *E. coli* isolates and fish feed have been claimed to be a source of antimicrobial-resistant bacteria. Based on the analysis of three studies, it has been observed that there is a current trend of an increase in ESBL-producing *E. coli* strains in aquatic environments [44,45,46]. The prevalence rates may vary in different regions and can be influenced by various factors, such as management practices, antibiotic usage, and the dissemination of resistant bacteria in the environment [43].

Livestock animals provide animal protein and are the source of meat and milk consumed by humans, since it is one of the major elements in the food chain [34,49]. Studies were also analyzed in which samples from food products were used, such as goat milk, ewe milk, fresh cheese, chicken meat, beef, and pork meat. A study was carried out in Spain by Ángel Alegría, and samples of goat milk, ewe milk, fresh cheese, chicken meat were studied. The detection of ESBL-producing strains in milk, dairy products, and chicken meat was detected in high prevalence [47]. Another study conducted in Portugal by Lurdes Clemente et al. used samples of beef, pork and broiler meat. It was found that 17.1% of all samples were contaminated, with the highest prevalence in poultry meat (30.3%). The prevalence in beef (11.8%) and pork (10.5%) samples was lower than in poultry, which is in accordance with the observations from other studies. This suggests that the high prevalence in poultry meat might be due to easier contamination along the food chain compared to beef and pork, and cross-contamination in flocks and slaughterhouse environments impacts the prevalence of ESBL Enterobacteriaceae in broiler chickens [48,50,51].

The presence of ESBL-producing bacteria in animals highlights the importance of the One Health approach in addressing the issue of antibiotic resistance. These studies demonstrate that food-producing animals have been identified as potential reservoirs and vectors of resistance genes [34,52].

## 3. Clonal Lineages of *Escherichia coli* in Livestock

In this review article, we gathered information from studies that investigated the presence and provide a general summary of the phenotypic and genotypic characteristics, as well as clonal genetic lineages (Multilocus Sequence Typing-MLST) of *E. coli* in food-producing animals (Table 2). MLST is an accurate and expansive molecular typing method which has been used and considered one of the most highly reproducible methods for typing and establishing clonal relationships between *E. coli* isolates. *E. coli* strains are assigned a sequence type (ST) with a numerical designation, according to standardized schemes [53]. There are several clones that have already been determined and found for *E. coli*, and there are clones that can be considered pandemic lineages and MDR (were classified as MDR, all strains that have resistance to at least three different classes) as is the case of ST131, ST69, ST95 and ST73 [53]. In this review we will analyze which are the most predominant clones that are found in different food-producing animals in different regions.

### 3.1. E. coli in Pigs: A Concerning Trend of Antimicrobial Resistance

Several studies have investigated the presence of *E. coli* in pigs. Nathaniel Storey et al. [54] conducted a study in the United Kingdom, where they sampled fecal samples from five different age classes of healthy pigs: weaners, gilts, farrowing, dry sows and grower/finishers. The study detected low resistance with only 7% considered MDR, compared with other conventional pig farms. They identified several sequence types (ST), with the most prevalent being ST744, ST44, ST88 and ST10. These STs showed resistance for all antimicrobial classes with the presence of detected antimicrobial-resistant (AMR) genes. Among the clonal lineages, different STs were identified, most of which were represented in only one host-livestock species. However, some STs were detected in multiple host species and were associated with extraintestinal pathogenic *E. coli* (ExPEC). For instance, ST10 (phylogroup A) has been linked to ExPEC pathotypes and has been reported in healthy pigs in the UK across multiple studies, indicating that this finding is not unusual [40,54,67,68,69]. ST744 clones were the most prevalent ST in this this UK pig study, and they have also been detected in seagull samples and wild-bird populations in Germany [70], suggesting wider transmission and recycling. Birds may be exposed to isolates from anthropogenic sources due to their scavenging activities [54]. Another study conducted in Brazil [55], analyzed eight samples of pig meat and showed high rates of resistance to β-lactams, with the identification of ESBL genes. The predominant ESBL genes found were blaCTX-M-55, blaCTX-M-15, blaCTX-M-2, and blaCMY-2 in pig meat. Two STs, ST410 and ST117, were detected in pigs, with ST117 also identified in samples collected from two other sources: humans and chicken meat. [55] This clone is often associated with AMR and is found in animals and human isolates, indicating a zoonotic profile [71,72]. *E. coli* carrying blaCTX-M genes from different sources seem to be related to the spread of internationally known clones that are distributed globally, including ST410 [55]. In Italy, Elisa Massella et al. [56] conducted a study analyzing 25 swine samples. The AMR profile was consistent with previous studies, with frequent detection of tetracycline, nalidixic acid, enrofloxacin, ampicillin, sulfamethoxazole, and trimethoprim resistance. However, the frequency of colistin and aminoglycoside resistance was low. Swine strains were associated with phylogroups A (32%), B1 (32%), C (32%), and E (4%). The most prevalent emerging ExPEC ST10 was detected, and other STs, such as ST641, ST3744, ST5759, ST100, ST20, ST206, ST871, ST410, ST7093, and ST88, were also found in the swine samples [56]. In Southwest Nigeria, Olusolabomi J. Adefioye et al. [57] conducted a study that involved phylogenetic characterization and multi-locus sequence typing of *E. coli* from food-producing animals. They examined a total of 19 pig isolates, and it was found that 42.1% of the *E. coli* strains were multi-drug resistant, with 13.3% being both multi-drug resistant and ESBL positive. All isolates were phenotypically resistant to clindamycin and penicillin. The ST131 cluster type, consisting of isolates from pigs, was considered the most prevalent and exhibited a high virulence potential, a broad-host range, and multi-drug resistant phenotypes. Some isolates of this clone harbored the *bla*CTX-M genes, making infections caused by them difficult to treat in both humans and food-producing animals [57,73]. Another study on *E. coli* isolated from pigs was conducted in Africa by Luria Leslie Founou et al. [59] in Cameroon and South Africa. The study analyzed *E. coli* isolates from nasal (*n* = 6) and rectal swabs (*n* = 5) of healthy pigs processed at abattoirs. All isolates were ESBL producers and exhibited high-resistance levels to ampicillin, cefuroxime, cefuroxime acetyl, third and fourth generation cephalosporins, and trimethoprim-sulfamethoxazole. The majority of isolates carried various aminoglycoside resistance genes, including *aph(3″)-Ib, aph(6)-1d, aadA5, and aadA1*. Plasmid-mediated quinolone resistance (PMQR) genes, including *qnrS1, aac(6′)Ib-cr*, and *oqxAB*, were also identified, along with mutations in the *gyrA* quinolone resistance-determining region (QRDR) genes. Resistance to trimethoprim and sulfamethoxazole was observed in all isolates, with each harboring at least one *sul* gene variant (*sul1* and *sul2*). Only one isolate carried the *mcr-1* gene, which encodes colistin resistance. The most prevalent clonal lineages were ST2144 and ST88, but the MDR-high-risk clone ST69 and ST10 were also detected [58]. *E. coli* ST10 was considered the main ST, along with ST131, which was identified in surgical site infections in the Central African Republic [74]. It is known that ESBL-producing *E. coli* can spread from one part of the world to another due to globalization and international travel. In this study, two ST2144 isolates originating from South Africa were identical and shared common ancestors with two Cameroonian isolates, ST940 and ST4450. An ST9440 isolate from South Africa shared common ancestors with three Cameroonian isolates, ST44, ST10, and ST226 [58,75]. Another study conducted in Switzerland by Claudine Fournier et al. [59] investigated the prevalence of *Enterobacterales* among pigs from a Swiss farm and determined the associated resistance mechanisms. They isolated and analyzed 81 fresh rectal swabs from healthy pigs and recovered a total of 38 *E. coli* isolates. A low rate of colistin-resistant isolates was found, and among the ESBL-producing *E. coli*, one major clone, ST10, was revealed. ST34 and ST744 were also identified and belonged to clonal complex CC10. Antibiotic resistance genes, such as *tetA* and *sul2*, were detected, and the high cephalosporin-resistance rate observed was mainly due to the spread of clonal strains [59]. In Austria, Tanja Bernreiter-Hofer et al. [60] studied 102 *E. col*i isolates from suckling and weaning pigs, characterizing their antimicrobial phenotypes. The majority of isolates showed a resistance to ampicillin (61.75%), tetracycline (58.81%), piperacillin (26.46%), sulfamethoxazole-trimethoprim (23.53%), cefotaxime (13.71%), and chloramphenicol (11.75%). The high-resistance rates to penicillin and tetracyclines were consistent with the results of previous studies, as these antibiotics are commonly associated with AMR in global pig production [1,60]. Overall, 36.27% of the isolates exhibited an MDR phenotype. The *bla*CTX gene family was the most prevalent, and the most common β-lactamase gene detected was *bla*TEM-1. The dominant phylogenetic groups were A (50.98%), followed by B1 (25.48%), C (8.81%), D (5.87%), B2 (3.91%), and F (1.95%). Whole Genome Sequencing (WGS) revealed that the most common sequence types were ST10 and ST100, although other sequence types, such as ST354, ST131, ST6404, ST6365, ST1112, ST1079, ST760, ST744, ST641, ST117, ST101, ST56, ST42, ST23, ST12008, ST12009, and ST12010, were also found. Regarding clonal lineage, ST131, which was found in this Austrian study, is described as a specific international high-risk clone with a wide distribution. This study represents the first report of ST131 in pigs of Austrian origin, emphasizing its close relationship between human and porcine populations [76]. *E. coli* ST10, an ancestral lineage occurring ubiquitously and comprising both commensal and pathogenic strains, is considered the dominant ST in swine populations in Northern Europe. It is associated with a broad-host range, including hospital- and community-acquired infections, and has recently been identified as primarily responsible for the spread of *mcr-4* [60,77,78]. Understanding the current pattern of antibiotics use in the pork industry is important for supporting and implementing measures to slow down the emergence of AMR in animal production and livestock environments. The classes of antibiotics used vary across countries and regions. Overall, penicillin and tetracyclines are the most commonly used antibiotics in pig production for prophylactic treatment, and they also show the highest rates of resistance. While the resistance to antibiotics may vary from region to region and based on sampling and AMR control methods used, high rates of resistance to tetracyclines and penicillin are consistently observed in multiple studies. For example, tetracycline resistance was reported in 58.81% of *E. coli* isolates in Austria and 61.75% among *E. coli* isolates in Cameroon [1,79]. Phylogenetic analyses showed some variation among the studies, but groups A and B1 were the most common in the analyzed studies.

The most frequent clones found were ST10, ST744, ST117, ST100, ST88, and ST131 (Figure 2), which are considered high-risk, zoonotic *E. coli* clones exhibiting both pathogenicity and multi-drug resistance. ST10 is associated with extraintestinal pathogenic *E. coli* and is known for its broad-host range, including hospital and community-acquired infections. It has been frequently detected in multiple studies and regions, including the United Kingdom, Germany, Italy, Austria, and the Central African Republic. ST131 has been identified in studies conducted in Brazil, Nigeria, and Austria and is known for its high-virulence potential, multi-drug resistance, and association with infections in humans and food-producing animals. Various other sequence types have been identified in different studies, indicating the genetic diversity and diverse clonal lineages of *E. coli* found in swine samples.

### 3.2. E. coli in Poultry: Multi-Drug Resistant Strains and Potential Reservoirs

Several studies have analyzed the genotypic, phenotypic characteristics and clonal lineages of *E. coli* in poultry. A study, conducted by Ama Szmolka et al. [61] in Hungary, isolated 90 *E. coli* strains from broiler chickens from various sources, including feces, caecum, bone marrow, and day-old chicks from three different slaughterhouses in North-Central Hungary. The predominant phylogroups were A (27%) and B1 (37%), and the most common sequence types (STs) were ST10, ST93, ST117, ST162, and ST155. ST10 is known as one of the most widespread multi-drug resistant (MDR) lineages in *E. coli* from animals, while ST93 includes ESBL and *mcr-1* expressing *E. coli* isolates from foods of animal origins. ST117, ST155, and ST162 were also identified, and they belong to different phylogroups [61,80,81]. Another study, conducted in Brazil by João Gabriel Material Soncini et al. [55], analyzed *E. coli* isolated from chicken meat and found that ESBL genes, including blaCTX-M-55, blaCTX-M-2, blaCTX-M-8, blaCTX-M-14, and blaCMY-2, were predominant. Certain STs, such as ST38, ST131, ST354, and ST1196, were detected in both human urine and chicken meat, indicating clonal dissemination between humans and poultry. ST117 was identified in samples from human urine, pork, and chicken, demonstrating clonal relatedness among isolates from different sources. ST38 was found in chicken meat and urine samples and has been linked to urinary tract infections [55,61,82]. Elisa Massella et al. [56] conducted a study in Italy and analyzed 25 poultry samples, from which, six *E. coli* strains were isolated. The study found significant resistance to quinolones, streptomycin, sulphonamides, tetracycline, ampicillin, and chloramphenicol in poultry samples, which can be attributed to antimicrobial usage in different sectors. Phylogroup C was the most common, and the prevalent STs were ST10 and ST117 [56,83]. In Southwest Nigeria, a study revealed that 38.9% of the *E. coli* strains isolated from poultry were multi-drug resistant, and 11.7% were ESBL-positive. High rates of resistance were observed for ampicillin, streptomycin, amoxicillin/clavulanate, cefotaxime, ticarcillin, ciprofloxacin, trimethoprim, and tetracycline. The most predominant phylogroup in poultry was A, and different sequence types, including ST131, ST410, and ST167, were detected among the multi-drug resistant ESBL isolates [83].

In Korea, a study conducted by Hyunsoo Kim et al. [62], collected samples from poultry farms, retail stores, slaughterhouses, and workers. The study found ESBL-*E. coli* rates of 6.8% in poultry, 0.9% in workers, 10.0% in chicken meat, and 14.3% in the environment. High-resistance rates were observed for nalidixic acid, ciprofloxacin, tetracycline, chloramphenicol, cotrimoxazole, and gentamicin. Various CTX-M-type ESBL genes were detected, with CTX-M-55, CTX-M-14, CTX-M-15, and CTX-M-27 being the most common types. Several other resistance genes associated with different classes of antibiotics were also identified. The studies conducted in different regions consistently demonstrate the presence of multi-drug resistant *E. coli* strains in poultry, indicating a widespread occurrence of antimicrobial resistance in poultry production. The sequence types identified in ESBL-*EC* isolates varied across the studies, with ST93 being the most common, followed by ST162, ST48, and ST115 (Figure 2). Clonal dissemination between poultry and humans was observed, with shared STs detected in both populations, highlighting the potential for zoonotic transmission and the exchange of multi-drug resistant strains [62]. *E. coli* strains belonging to the pathogenic group B2, which possess multiple virulence traits associated with various infections, are of particular concern.

Poultry and poultry products have been identified as potential reservoirs for *E. coli* in humans due to the close genetic relationship between avian pathogenic *E. coli* and human pathogenic strains [84]. The increased use of antimicrobial agents in poultry production is considered a contributing factor to the widespread occurrence of antibiotic resistance in poultry [83,85]. In summary, these studies provide valuable insights into the prevalence of multi-drug resistant *E. coli* strains in poultry, and highlight the need for measures to mitigate the spread of antimicrobial resistance between animals and humans [77].

### 3.3. E. coli in Rabbits: High Levels of Resistance to β-Lactamases and Colistin

Regarding studies conducted in rabbits, there are still few studies that establish a connection between antibiotic resistance and clonal strains of *E. coli* obtained from production rabbits. A study conducted in Italy analyzed 14 samples obtained from rabbits, of which, 10 samples were identified as *E. coli* with a high-resistant phenotype to common antibiotics such as tetracycline, sulphonamides, streptomycin, and ampicillin. Interestingly, rabbits were mostly associated with colistin resistance. Phylogroup B1 was the primary association with rabbits, and the most detected sequence types were ST20, followed by ST40 [56]. In China, Xiaonan Zhao et al. [63] conducted a study on the molecular characterization of antimicrobial resistance in *E. coli* from rabbit farms. They collected 60 fecal samples from diarrhea-stricken rabbits on farms and isolated a total of 55 *E. coli* strains. The study showed high resistance to tetracycline and ampicillin, and 50.9% of the isolates were considered multi-drug resistant strains. These resistances have also been observed in other studies involving different production animals, but compared to the above study, there was concordance in the resistance panorama for tetracyclines and ampicillin [56,63]. In this study, this happens is because tetracycline and ampicillin have been widely used to control and prevent rabbit diseases in the Tai’an area of China. The most frequently detected genes were β-lactamase genes, *bla*TEM (98.2%) and *bla*CTX-M (94.5%), followed by *sul2* (58.18%), *tetB* (9.1%), *qnrS* (5.5%), and *aac(6)-Ib-cr* (7.5%). Various STs were identified, with ST302 (40.0%) being the most prevalent, followed by ST370 (21.8%). ST40 has been previously reported in studies related to infections associated with rabbits and is the most common ST found in rabbits in the Italian study [63]. In Portugal, a study focused on three *E. coli* isolates obtained from the intestinal content of necropsied meat rabbits collected from two commercial farms. These isolates were found to be resistant to colistin, and the *mcr-1* gene was detected in the isolates. This discovery was reported as the first in Portugal, although it had already been detected in Italy in 2018 [64,86]. The rabbit industry is mostly associated with colistin resistance, as described in several articles, due to various factors, including the fragile intestinal microbiota of rabbits, intestinal health problems, and their inability to adjust to a diet with a high nutritional density or the stress inherent in intensive farming, as seen in poultry, cattle, and pigs [64,86,87]. The identification of colistin resistance was also associated with rabbit meat, highlighting the possible involvement of the rabbit breeding system in colistin resistance and its dissemination through the food-chain [56].

The different studies conducted on rabbits have identified some common findings. All the studies reported the presence of antibiotic-resistant strains of *E. coli*, with high levels of resistance observed against antibiotics such as tetracycline, sulphonamides, streptomycin, and ampicillin. Phylogroup B1 was predominantly associated with rabbits in the studies. Multiple STs [56] were identified, including ST20, ST40, ST302, and ST1431. Some of these strains have also been identified in other food-producing animals, such as turkey meat samples from Germany. ST40 has been previously associated with the dissemination of CTX-M-1 and OXA-48 in humans, companion animals, and poultry, suggesting potential clonal dissemination and zoonotic transmission of *E. coli* strains [64,88]. The studies also consistently identified the presence of β-lactamases genes, particularly *bla*TEM and *bla*CTX-M.

### 3.4. E. coli in Cattle: Potential Cross-Species Transmission

Among the various food-producing animals, cattle are also part of this economic sector; therefore, several studies have been conducted and analyzed regarding samples from cattle. In an Italian study, samples from dairy cattle were analyzed, and phylogroup B1 was found to be the most prevalent (36%). The most prevalent sequence types (STs) were ST69, ST10, and ST58. However, ST10 has also been identified in other food-producing animals such as pigs and poultry. The high frequency of resistance to sulfonamide-ampicillin-tetracycline was consistent with previously reported AMR profiles in Europe [56]. In a study conducted in Nigeria, high levels of antibiotic resistance and multi-drug resistant *E. coli* isolates were detected in cattle. The isolates mainly belonged to the less common phylogroup D, and ST131 and ST405 were identified. ST131 was also found in samples from humans, pigs, goats, poultry, and beef. In the same study, beef samples collected from market retailers also showed the presence of ST131, followed by ST405. Beef samples were mainly clustered into phylogroups A and B1, while phylogroups B2 and D have clinical and epidemiological implications as they could potentially be transferred along the food chain [57,73]. In China, Yu-Long Zhang et al. [89] conducted a study where 222 *E. coli* isolates were recovered from 230 cloacal swabs of healthy beef cattle. Among these isolates, 45.9% exhibited the ESBL phenotype. The isolates showed phenotypic resistance to commonly used β-lactam drugs, particularly penicillin derivatives and third-generation cephalosporins, which are commonly used in China. A high prevalence of CTX-M and SHV genes was observed. In terms of phylogenetic groups, most isolates belonged to phylogroup A (*n* = 44), followed by B1 (*n* = 30), B2 (*n* = 9), and D (*n* = 5). Different sequence types were identified, with the most common being sequence type 398 (associated with *E. coli* infections) and sequence type 7130 [89]. It has been described that *E. coli* from cattle farms exhibit greater sequence type diversity compared to those from pig farms [68]. Another study, conducted by Jonathan Massé et al. [65], focused on the antimicrobial resistance of *E. coli* on dairy farms in Canada. They collected fecal samples from 101 commercial dairy farms. The most frequent resistance observed on dairy farms was against ampicillin (98%), followed by ceftriaxone (90%), sulfisoxazole (88%), ceftiofur (84%), and tetracycline (80%). The predominant AMR genes detected were *sul2* (72%), *strA/strB* (65%), *tet(A)* (53%), *aph(3′)-1a* (47%), and *bla*CTX-M, which was detected in 42% of all isolates. A total of 14 clonal lineages were identified in this study, with clonal lineage I ST10 (phylogroup A) and clonal lineage IX ST88 (phylogroup C) being the most widespread lineages. Some clones were found on farms that were geographically far apart, suggesting the dispersal of resistant clones between dairy farms. The transport of live animals between farms could be one explanation for the spread of these clones. The presence of clonal lineage ST117, which is associated with poultry, was also found in this study [65]. These studies demonstrate that resistance to broad-spectrum β-lactams and fluoroquinolones, as well as the presence of relevant-resistant genes, are widespread among *E. coli* populations on dairy farms. However, in 2019, a new regulation named “Categorization of Antimicrobial Drugs Based on Importance in Human Medicine” was implemented in Québec, Canada, by the Government of Canada. This regulation aims to assist in the microbiological safety assessment of veterinary antibiotics in pre- and post-market evaluations, with a focus on human health [65,90]. In summary, the studies conducted in Italy and Nigeria reported the prevalence of phylogroup B1 in *E. coli* isolates from cattle. ST10 was found in multiple studies involving different food-producing animals, including rabbits, pigs, and cattle, indicating potential cross-species transmission or a common environmental reservoir for this sequence type. The Canadian study on dairy farms identified ST10 and ST88 as the most widespread lineages, suggesting potential dissemination through animal transport. The Italian study on dairy cattle reported a high frequency of resistance to sulfonamide–ampicillin–tetracycline, consistent with previously reported AMR profiles in Europe. A high prevalence of the ESBL phenotype and the presence of CTX-M and SHV genes were consistently observed in all studies.

### 3.5. E. coli in Small Ruminants: Widespread Dissemination of Antibiotic Resistance

The last food-producing animals studied were small ruminants, specifically sheep and goats, and both studies were conducted in Nigeria. In terms of the multi-resistant profile of *E. coli* isolates, a resistance rate to the tested antibiotics was found to be 13.3% in goats and 1.7% in sheep, indicating a high level of resistance. Sequence type ST131 was identified in strains from both goats and sheep, and this sequence type is also frequently found in other sources such as humans, pigs, cattle, poultry, and beef [57]. This review focuses on investigating *E. coli* isolates among a variety of food-producing animals that are considered potential risks for the dissemination of ESBL-producing *E. coli*. This dissemination could have significant implications for human and animal health [21,89]. In an Italian study, fecal samples from goats were analyzed, and a resistance to colistin and tetracycline was found. The presence of the *gadW* gene, associated with multi-drug resistance, suggests that goats can harbor bacteria resistant to multiple classes of antibiotics. The occurrence of *E. coli* ST675 was also detected [66]. In China, a study conducted by Xueliang Zhao et al. [20] aimed to identify and characterize the resistance profiles of 67 ESBL-producing *E. coli* isolates from sheep in northwest China. Resistance to ceftazidime, ceftiofur, ceftriaxone, and cefixime was observed in 100% of the isolates, and 38.8% of the isolates were resistant to cefepime. High rates of resistance to tetracyclines (80.6%), mequindox (76.1%), enrofloxacin (76.1%), ampicillin (70.1%), spectinomycin (68.7%), gentamicin (55.2%), and colistin (29.9%) were also reported in this study. Different STs were found, but most of the isolates harbored ST10, ST23, ST58, ST162, ST167, ST361, ST602, and ST1137. The emergence of *E. coli* isolates resistant to third and fourth-generation cephalosporin and colistin in sheep is a cause for concern, as these antibiotics are not globally authorized or approved for use in sheep. Other studies conducted in Portugal [91] and the USA [92] have confirmed the presence of multi-drug resistant ESBL-producing *E. coli* in sheep. ST10, ST23, and ST58 were the most prevalent types identified in terms of clonal relationship [20]. Several studies conducted in different regions of the world, including Nigeria, Italy, China, Portugal and the USA, have collectively highlighted the concerning presence of antibiotic-resistant *E. coli* isolates in small ruminants, such as sheep and goats. These studied have identified the emergence of multi-resistance profiles, including to antibiotics such as colistin, tetracycline, third and fourth-generation cephalosporins. The detection of similar resistance patterns and STs across these studies suggests a potential correlation and widespread dissemination of antibiotic-resistant *E. coli* strains among different geographical regions.

The application of next-generation sequencing technologies has revealed the spread of multi-drug resistant *E. coli* strains, and some clonal strains have been found to be disseminated in environments associated with livestock. These strains are associated with pathogenic clonal lineages that successfully combine multi-drug resistance and high virulence or emerge as globally disseminated pathogens [93]. Analysis of the studies in Table 2 has shown that the diversity of clonal lineages of the strains depends on the proximity of animals and even humans, even if some of these studies did not involve close contact with farm animals [94]. Clonal group A, belonging to the D phylogroup and sequence type ST131, is among the most common and widely spread clones associated with antimicrobial resistance in *E. coli* and extended-spectrum beta-lactamases (ESBLs) [94,95]. The strains of animal origin are associated with the antimicrobial-resistance problem; however, an even bigger problem arises when, these strains that are present in the animals, manage to exceed this limit and pass through various stages of the food chain, potentially disseminating in humans. To combat this, appropriate hygiene practices and better control of the use of antimicrobial agents should be implemented to limit the dissemination of multi-drug resistant organisms within the community [84,94].

## 4. Methods

A literature search was performed in the Web of Science and in PubMed. Google Scholar was used to identify the relevant grey literature. The search terms used were “antibiotic resistance”, “antimicrobials in food-producing animals”, “livestock animals”, “*Esherichia coli*”, ”antibiotics used in livestock”, ”food-producing animals”, “phylogenetic groups in *E. coli*”, “ESBL”, and “genetic lineages”. A restriction about publication type or year were applied, for example, preference was given to articles published from 2019 to the present. The search was conducted in April 2023. The tables were created using consistent criteria across all the compiled studies. In the event of variation in the methodology, these differences were explicitly indicated within the table. The literature review involved a comprehensive comparative analysis and qualitative synthesis. The goal was to identify common patterns and concepts across multiple studies, aiming to develop a deeper understanding of the research topic. Additionally, the review aimed to compare and contrast the findings from different studies conducted in various regions of the world, with the goal of identifying both similarities and differences. The problem of antibiotic resistance is general, in both human and animal medicine. However, the emergence of resistant strains to the most used antibiotics has generated a huge concern, since there are no longer therapeutic options in both human medicine and veterinary medicine. It was also noticed that one of the main causes of the evolution of these strains would be the production animals since they are part of the food chain and are in contact with different environments. This review summarized studies on the antimicrobial resistance and the genetic lineages found in livestock animals all over the world. The prevalence of *E. coli-*ESBL strains was also explored, to understand the epidemiological panorama and the prevalence found in *E. coli* strains isolated from a variety of samples of food-producing animals.

## 5. Conclusions

The intense use of antibiotics in livestock production causes the emergence and rapid spread of MDR and ESBLs in *E. coli*, and may lead to difficult-to-treat infections and constitutes a major reservoir of resistance determinants to most families of antimicrobial agents. High levels of antimicrobial usage were found in the different food-producing animals, especially tetracyclines, fluoroquinolones, β-lactams and aminoglycosides and suggested widespread environmental AMR pollution and, consequently, a possible transmission path of potential multi-resistant pathogens to humans via the food chain. In fact, different clonal lineages were found in pigs, poultry, cattle, rabbits, goats, and sheep; however, it was found that some clonal lineages such as ST131, ST117, among others, are disseminated in different ecological niches. The clonal lineage ST131 is considered a worldwide pandemic clone that has been found in several studies. Therefore, better implementation measures, restricting the use of antibiotics in farm animals, and control and treatment strategies are needed to reduce the emergence and spread of AMR with a scope in the food safety perspective.

## Figures and Tables

**Figure 1 antibiotics-12-01061-f001:**
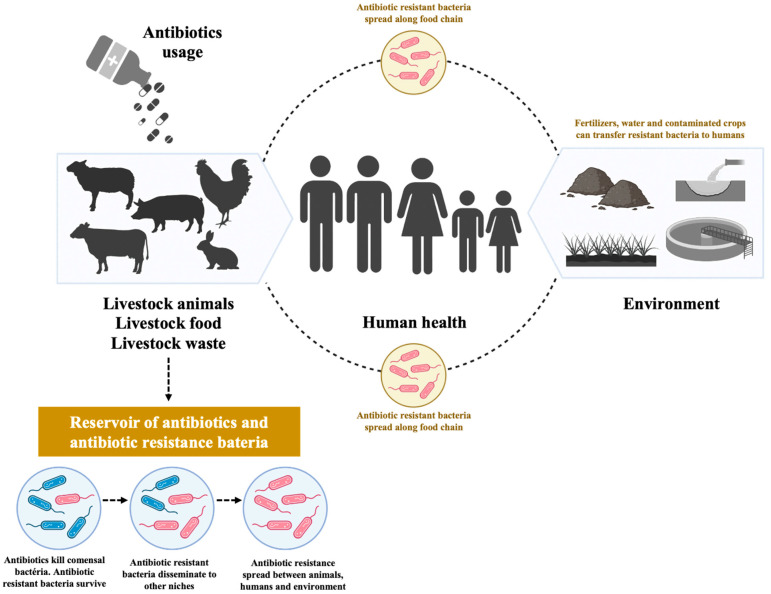
Antimicrobial resistance in livestock, humans, and environment.

**Figure 2 antibiotics-12-01061-f002:**
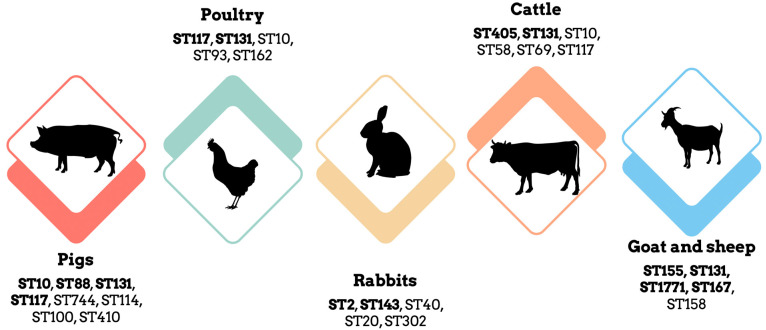
Distribution of the clonal lineages in livestock animals (pigs, poultry, rabbits, cattle, goat, and sheep), the most frequent clonal lineages of each food-producing animals are in bold.

**Table 1 antibiotics-12-01061-t001:** Prevalence of ESBL-producing *E. coli* in livestock animals.

Samples	Location	Data of Isolation Samples	Food-Producing Animals	Total Samples Collected	Method Used for ESBL Detected	Number of Samples Detected with ESBL—*E. coli* Positive	Prevalence ESBL—*E. coli* Positive (%)	Reference
Fecal	Guadeloupe		Poultry	216	Double-disk synergy test	6	14.7	[35]
		2018–2019	Pigs			9	7.3	
			Pigs			11	35.3	
Fecal/cacecal intestinal content	Italy	2016–2017	Pigs	445	Enrichment broth+CTX/double-disk synergy test	120	27	[36]
			Cattle			120	27	
			Poultry			194	43.6	
Fecal	Ghana	2015	Poultry	140	Enrichment broth+CTX/double-disk synergy test	41	29	[37]
Fecal	Belgium	2017–2018	Pigs	798	Double-disk synergy test	37	48	[38]
			Broilers			45	58.4	
Fecal	Hungary	ND	Poultry	124	Double-disk synergy test	39	34.2	[41]
			Porcine	100		72	72	
Food (Farms)	Spain	ND	Chicken meat	10	Chromagar ESBL	2	15.4	[47]
			Goat milk	68		3	23	
			Fresh cheese	20		3	23	
			Ewe’s milk	10		5	38.5	
Caecal	Nigeria	2019	Spent layers	50	Double-disk synergy test	4	8	[39]
			Broilers	304		33	10.8	
Boot/rectal (cattle)	Reunion island	2015–2018	Rabbits	39	Double-disk synergy	0	0	[42]
			Cattle	124		2	8.3	
			Sheep and Goats	50		3	18.5	
			Pig	177		33	50	
			Poultry	176		50	70	
Fecal	Malaysia	ND	Rabbits	100	PCR	0	0	[43]
			Swine	100		0	0	
			Poultry	200		11	5.5	
Fecal	South Korea	2018	Chickens	32	Chromagar ESBL+ Double-disk synergy test	4	7	[40]
			Pigs	59		41	69.5	
			Cattle	34		32	94.1	
Food (meat)	Portugal	2016–2017	Pork	220	Double-disk synergy test	23	10.5	[48]
			Beef	220		26	11.8	
			Broiler	198		60	30.3	
Fish samples-site	India	2019	Fish	66	Double-disk synergy test	54	81.8	[46]
Shrimp, water, and sediment	India	2018–2020	Aquaculture farms	261	BD PhoenixTM M50 automated system	14	5.4	[44]
Fish	Nigeria	ND	Fish farms	90	Double-disk synergy test	54	60	[45]

ND—Not determined.

**Table 2 antibiotics-12-01061-t002:** Phenotypic, genotypic characteristics and clonal genetic lineages (MLST) of *E. coli* in food-producing animals.

Animal	Samples	Location	Data of Isolation Samples	Most Prevalent Phylogroup	Clonal Lineages (MLST)	AMR Phenotypes	AMR Genotypes	MDR (%)	References
Pigs	Faecal	United Kingdon	2017–2018	ND	ST44, ST88, ST10, ST744, ST58, ST117, ST48, ST2721	Aminoglycosides, β-lactams, fluoroquinolones, sulphonamides, tetracyclines	*aadA5, strA, strB* *aph (6)-Ib, blaTEM-1b, mphA, sul, sul2, tetAB, tetB, dfrA5*	7	[54]
	Meat	Brazil	2016–2019	ND	ST410, ST117	Aminoglycosides, β-lactams, trimethoprim, phenicols, tetracyclines, macrolides, sulfonamides, quinolones, lincosamides, fosfomycin	*strA and strB, aadA1*, *dfrA17, dfrA1, tet(A), tet(B)*, *sul1, sul2*, *of blaCTX-M-55*, *blaCTX-M-15*, *blaCTX-M-2*, *blaCMY-2*	ND	[55]
	Feacal	Italy	2010–2018	A (32%), B1 (32%),C (32%), E(4%)	ST10, ST641, ST3744,ST575,ST100, ST20, ST206, ST871, ST410, ST7093, ST88	Streptomycin, chloramphenicol, sulfisoxazole, trimethoprim/sulfamethoxazole, tetracycline, nalidixic acid; enrofloxacin, colistin	*blaTEM-1b, aadA1, strA, strB, sul1, dfrA1 tetA, tetB*	24	[56]
	Feacal	Nigeria	2015–2016	D, B2, B1, A	ST131, ST2348	Clindamycin, penicillin, ceftazidime, tobramycin, cefazolin, enrofloxacin, levofloxacin, sulfamethoxazole/trimethoprim, kanamycin, cefuroxime, piperacillin/tazobactam, ampicillin, cefalexin, streptomycin, doxycycline, neomycin, spectinomycin, amoxicillin/clavulanate, sulfamethoxazole, ampicillin/sulbactam, cefotaxime, ticarcillin, ciprofloxacin, trimethoprim, tetracycline	*blaCTX-M-15, blaCTX-M-1, blaCTX-2, blaCTX-M-9*	42.1	[57]
	Nasal and rectal	Cameroon and South Africa	2016	A (45%), B1 (28%) and C (18%), D (9%)	ST88, ST2144, ST10, ST69, ST226, ST944, ST4450, ST44	ampicillin, cefuroxime, cefuroxime acetyl, cefotaxime, ceftazidime, cefepime, trimethoprim–sulfamethoxazole	*blaCTX-M-15, blaTEM-1B, blaTEM-141, blaTEM-206, aph(3″)-Ib, aph(6)-1d, aadA5, aadA1, qnrS1, aac(6′)Ib-cr, oqxAB, gyrA, sul2, sul1, dfrA17, dfrA14*	18.18	[58]
	Rectal	Switzerland	ND	A (57.9%), B (34.3%), C (2.6%), B1 (31.5%), B2 (2.6%)	ST10, ST34, ST744	Tetracycline, sulfonamides, gentamicin, tobramycin; kanamycin	*blaCTX-M-1*	ND	[59]
	gut-associated	Austria	ND	A (50.98%), B1 (25.48%), C (8.81%), D (5.87%), B2 (3.91%), F (1.95%), E (0.97%), G (0.97%)	ST10, ST100, ST354, ST131, ST6404,ST636,ST1112,ST107,ST760, ST744, ST641, ST117, ST101, ST56, ST42, ST23, ST12008, ST12009, ST12010	Ampicillin, tetracycline, piperacillin, sulfamethoxazole–trimethoprim, cefotaxime, chloramphenicol, ceftazidime, cefepime, gentamicin, fluoroquinolone, aztreonam, tobramycin, and fosfomycin, colistin	*blaCTX, blaCMY-2, blaTEM-1, blaCTX M-1, mcr-1.1, gyrA, parC*	36.27	[60]
Poultry	Faecal, caecum and bone marrow	Hungary	2016–2018	A (27%) B1 (37%)	ST10, ST93, ST117, ST162, ST155, ST8702, ST10088	Aminoglycosides, β-lactams, fluoroquinolone, sulphonamides, tetracyclines, trimethoprim.	*blaTEM-1, tet (A), aadA1, aph (3″)-Ib, aph (6)-Id, sul2*	ND	[61]
	Meat	Brazil	2016–2019	ND	ST38, ST131, ST354, ST1196, ST117	Aminoglycosides, β-lactams, trimethoprim, phenicols, tetracyclines, macrolides, sulfonamides, quinolones, lincosamides, fosfomycin	*strA and strB, aadA1, dfrA17, dfrA, tet(A), tet(B), sul1, sul2*	ND	[55]
	Food	Italy	2010–2018	A (8%), B1 (24%), B2 (4%), C (32%), E (8%) F (20%)	ST23, ST101, ST359, ST131, ST117, ST744, ST57, ST48, ST162, ST10, ST155, ST2614, ST297, ST93, ST69, ST1286	Gentamicin, Streptomycin, chloramphenicol, sulfisoxazole, trimethoprim/sulfamethoxazole, tetracycline, nalidixic acid; enrofloxacin	*blaTEM-1b, aadA1, strA, strB, sul1, dfrA1 tetA, tetB*	64	[56]
	Feacal	Nigeria	2015–2016	B2, B1, A	ST131, ST156, ST167, ST410, ST1056	Clindamycin, penicillin, ceftazidime, tobramycin, cefazolin, enrofloxacin, levofloxacin, sulfamethoxazole/trimethoprim, kanamycin, cefuroxime, piperacillin/tazobactam, ampicillin, cefalexin, streptomycin, doxycycline, neomycin, spectinomycin, amoxicillin/clavulanate, sulfamethoxazole, ampicillin/sulbactam, cefotaxime, ticarcillin, ciprofloxacin, trimethoprim, and tetracycline	*blaCTX-M-15, blaCTX-M-1, blaCTX-2, blaCTX-M-9*	38.9	[57]
	Chicken meat and environment	Korea	2019	ND	ST93, ST131, ST48, ST57, ST69, ST88, ST115, ST117, ST162, ST297, ST362, ST457, ST770, ST919, ST1011, ST143, ST1485, ST163, ST165, ST2179, ST2334, ST278, ST2792, ST328, ST3941,ST455, ST6779	nalidixic acid, ciprofloxacin, tetracycline, chloramphenicol, cotrimoxazole, gentamicin	*bla, str, aad, aac, aph, mph, aac(6′)Ib-cr, Qnr, dfr, sul, tet, cat, fos, ARR-3, Inu,* *CTX-M-55, CTX-M-14, CTX-M-, CTX-M-15, CTX-M-65, CTX-M-27 and CTX-M-61*	ND	[62]
Rabbits	Food	Italy	2010–2018	B1 (86.96%), B2 (4.35%), E (8%), D (8.70%)	ST40, ST20, ST1611, ST297, ST533, ST20, ST129, ST706, ST906, ST351, ST501, ST224, ST111, ST539, ST1431, ST491, ST1727	Gentamicin, Streptomycin, chloramphenicol, sulfisoxazole, trimethoprim/sulfamethoxazole, tetracycline, nalidixic acid; enrofloxacin, colistin	*blaTEM-1b, aadA1, strA, strB, sul1*, *dfrA1 tetA, tetB*	95.65	[56]
	Feacal	China	2016	ND	ST302, ST468, ST370, ST87, ST314, ST370, ST636, ST2, ST24, ST88, ST353, ST370, ST461, ST731, ST73	Tetracycline, ampicillin, chloramphenicol, ciprofloxacin gentamicin, nalidixic acid, trimethoprim/sulfamethoxazole	*blaTEM, blaCTX-M, sul2, tetB qnrS, aac(6)-Ib-cr*	50.9	[63]
	Intestinal content	Portugal	ND	A, B1	ST206, ST1589, ST1431, ST2, ST4	Colistin	*mcr-1*	ND	[64]
Cattle	Food	Italy	2010–2018	A (32%), B1 (36%) B2 (8%), C (16%), D (4%), E (4%)	ST1510, ST398, ST10, ST583, ST1303, ST58, ST155, ST69, ST278, ST1091, ST731, ST2328, ST216, ST1125	Gentamicin, Streptomycin, chloramphenicol, sulfisoxazole, trimethoprim/sulfamethoxazole, tetracycline, nalidixic acid; enrofloxacin, colistin, ceftiofur, ceftazidime	*blaTEM-1b, aadA1, strA, strB, sul1, dfrA1 tetA, tetB*	24	[56]
	Beef	Nigeria	2015–2016	D, B2, B1, A	ST58, ST131, ST405	Clindamycin, penicillin, ceftazidime, tobramycin, cefazolin, enrofloxacin, levofloxacin, sulfamethoxazole/trimethoprim, kanamycin, cefuroxime, piperacillin/tazobactam, ampicillin, cefalexin, streptomycin, doxycycline, neomycin, spectinomycin, amoxicillin/clavulanate, sulfamethoxazole, ampicillin/sulbactam, cefotaxime, ticarcillin, ciprofloxacin, trimethoprim, and tetracycline	*blaCTX-M-15, blaCTX-M-1, blaCTX-2, blaCTX-M-9*	22.6	[57]
	Feacal		2015–2016	D, B2, B1, A	ST131, ST405			33.3	
	Beef cattle/Cloacal	China	2016	A (50%), B1 (34%), B2 (10.22%), D (5.6%)	ST398, ST7130, ST297, ST48, ST4977, ST202	β-lactam, penicillin derivatives and third generation cephalosporins	*blaCTX-M, blaTEM, blaSHV*	ND	[63]
Dairy farms	Feacal	Canada	ND	A (33.7%), E (4.6%), D (5.8%), F (3.4%), G (4.6%), C (18.6%), B1(29%)	ST10, ST88, ST744, ST4981, ST2500, ST34, ST48, ST11813, ST5708, ST408, ST540, ST1204, ST219, ST3018, ST2449, ST38, ST69, ST967, ST648, ST1163, ST657, ST117, ST783, ST21, ST4559, ST162, ST2522, ST172, ST345, ST297, ST155, ST683	Ampicillin, ceftriaxone, sulfisoxazole, ceftiofur, tetracycline, ciprofloxacin, danofloxacin, enrofloxacin, and nalidixic acid, azithromycin, gentamicin	*sul2*, *strA/strB, tet(A), aph(3′)-1a, blaCTX-M, blaCMY-2*, *ampC, qnrS1, blaCTX-M-15, blaCTX-M-1, gyrA, parC, parE*	ND	[65]
Goats	Feacal	Nigeria	2015–2016	B2, B1, A	ST131, ST155, ST167, ST406, ST1771	Clindamycin, penicillin, ceftazidime, tobramycin, cefazolin, enrofloxacin, levofloxacin, sulfamethoxazole/trimethoprim, kanamycin, cefuroxime, piperacillin/tazobactam, ampicillin, cefalexin, streptomycin, doxycycline, neomycin, spectinomycin, amoxicillin/clavulanate, sulfamethoxazole, ampicillin/sulbactam, cefotaxime, ticarcillin, ciprofloxacin, trimethoprim, and tetracycline	*blaCTX-M-15, blaCTX-M-1, blaCTX-2, blaCTX-M-9, blaCTX-M-11*	50	[57]
	Feacal	Italy	2019	ND	ST675	Colistin, tetracycline	*KpnE, KpnF, acrD, baeR, baeS, cpxA, tolC, soxS, soxR, marA, ampC, ampC 1, acrA, acrB, acrE, acrF, acrR, acrS, CRP, emrE, evgA, evgS, gadX, gadW*	ND	[66]
Sheeps	Feacal	Nigeria	2015–2016	B1	ST58, ST131, ST155, ST156, ST167, ST405, ST406, ST1056, ST1771, ST2348	Clindamycin, penicillin, ceftazidime, tobramycin, cefazolin, enrofloxacin, levofloxacin, sulfamethoxazole/trimethoprim, kanamycin, cefuroxime, piperacillin/tazobactam, ampicillin, cefalexin, streptomycin, doxycycline, neomycin, spectinomycin, amoxicillin/clavulanate, sulfamethoxazole, ampicillin/sulbactam, cefotaxime, ticarcillin, ciprofloxacin, trimethoprim, and tetracycline	ND	5.5	[57]
	Cloacal swabbing	China	2019–2020	B1(70.1%), B2 (1.5%), C (20.9%), E (1.5%), F (1.5%)	ST10, ST23, ST58, ST162, ST167, ST361, ST602,ST1137	Ceftazidime, ceftiofur, ceftriaxone, cefixime (third generation), cefepime (fourth generation), sulfisoxazole, florfenicol, tetracyclines, mequindox, enrofloxacin, ampicillin, spectinomycin, gentamicin, colistin	*blaCTX-M, blaTEM, blaOXA. blaSHV, blaCMY, blaKPC*	6.7	[20]

ND—Not determined; MLST—Multilocus sequence typing; ST—Sequence type; AMR—Antimicrobial resistance; MDR—Multi-drug resistance.

## Data Availability

No new data were created or analyzed in this study. Data sharing is not applicable to this article.

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
