# Peer review of "Antimicrobial Resistance and Clonal Lineages of Escherichia coli from Food-Producing Animals"

_antibiotics, 2023, doi:10.3390/antibiotics12061061_

Round 1
Reviewer 1 Report
Dear author,
The language of paper needs strong revision. In some parts it has several problems that made it difficult to understand. Moreover, in the first part of the paper there is no logical order for information that presented. Furthemore, your search was limited to some countries. Did it have specific reason?

Language of the paper eeds revision. Some parts were shown by highlights.
Author Response
- The language of paper needs strong revision. In some parts it has several problems that made it difficult to understand. Moreover, in the first part of the paper there is no logical order for information that presented. Furthemore, your search was limited to some countries. Did it have specific reason?
Authors’response:
All the suggested changes were implemented in the revised manuscript.
We did a search so that only current articles were used, and the data was current, from 2019 until now. In the research we did, these were the articles that appeared and that we selected so that even part of the methodology used was used. We tried to put examples from various parts of the world so that it was possible to see which panamora we are currently in. However, more information has been added in the revised manuscript.
Reviewer 2 Report
This work "Antimicrobial resistance prevalence and clonal lineages of Escherichia coli from food-producing animals "mainly focused on the resistance profiles of E. coli and ESBL-E. coli in food-production animals. I think that a sufficient and current review in relation to the antimicrobial resistance of Escherichia coli from food-producing animals is performed, and the study is a useful one.
Author Response
We appreciate the reviewer's comments.
Reviewer 3 Report
- For the manuscript (including the title), can author confirm that “clonal lineage” is the technically accurate terminology used? as the discussion was mainly only focus on comparison of sequence type but not other e.g. SNP alignment etc. Can author confirm “sequence type” and “clonal lineage” are not different?
- For the sections on prevalence of E. coli/MDR rate, suggest authors to describe the methodology briefly for the review, e.g. sources of articles reviewed, keywords used for the searching, selection/exclusion criteria for articles (e.g. period), analysis approach for the secondary analysis. The description of secondary analysis is important as different ESBL E. coli methodology (e.g. enrichment method/direct culture/PCR or other method) or analysis approach (e.g. definition of MDR i.e. resistant to at least 3 classes of antimicrobials) would need to be taken into consideration when comparison of prevalence/MDR rate.
- would suggest author to include a key message for each session,. For instance, instead of “antimicrobials use in animal production” can author highlight message eg “antimicrobials use in animal production showing an increasing/tapering trend?; Though varied prevalence of ESBL E. coli between geographical regions, certain ST has been found to be predominating sequence type among the livestock animals ”etc. Quite large portion of the current text includes restating of statistics, it can be quite difficult to follow for key messages of the manuscript. Hence including key messages as a subtitle for each session would definitely help reader to grasp the key messages of the review. For authors’ consideration
- Authors in the text mentioned several of background on virulence and pathogenicity of E. coli as justification of this review. E. coli are generally not pathogenic; while concern of AMR is “valid” only when a microorganism causes infections which cannot be treated using the antimicrobials e.g. more severe/invasive salmonellosis requiring treatment. Hence, I believe that a rationale of that E. coli is still studied, would be that it an indicator organism of warm-blooded animals which would allow us to better understand transmission of AMR within livestock or between compartments? (e.g. between different animal species; or between animal and humans; or between animal and environment etc). To this, I would suggest authors to discuss about location of the AMR gene detected (either chromosomally or plasmid-mediated) in E. coli, e.g. to further discuss potential roles of E. coli in “spreading” AMR determinants to other more pathogenic bacteria through horizontal transfer.
Would suggest author to consider the below reference.
FAO. 2019. Monitoring and surveillance of antimicrobial resistance in bacteria from healthy food
animals intended for consumption. Regional Antimicrobial Resistance Monitoring and Surveillance Guidelines – Volume 1. Bangkok
- The para with the first line 164 needs rephrasing for more clarity. For example, I don’t quite get the below sentences.
· Several studies in Table 1 were carried out all over the world regarding the inclusion of ESBLs-producing E. coli in a diversity of animals as in cattle, swine’s and poultry isolated from fecal samples or food samples more specifically in meat product
· In Guadeloupe, a study made by Gaëlle Gruel et al. in pigs, beef cattle and poul-169 try showed moderate (of what? prevalence?) ESBL-E. coli in pig and beef cattle production, in this island (where?) an anti-170 microbial resistance plan (is author refers to antimicrobial stewardship programme?) on the use of antimicrobials in veterinary has been implemented 171 and the moderate ESBL resistance prevalence probably reflect its effectiveness in the ra-172 tional use of antimicrobials, in broilers, a high frequency of ESBL resistance was observed.
- For Table 1. The header of the 6th column needs more clarity, is author refers to “number of sample detected with ESBL-E. coli”. Can author add a column for the table 1 to indicate the method of ESBL e.g. enrichment method, PCR method, Vitek etc, for readers’ consideration when reference for comparison. As it might not be fair comparison without considering the method used.
- For Table 2. There should be information on isolation year. Suggest author to define MDR, as different papers might refer MDR differently. What means by ‘-‘? Is it refer to that data not available or 0% prevalence—both are different. Authors might want to consider further differentiate the genotype (chromosomally-mediated or plasmid-mediated) as per comment SN4.
- As the conclusion section states “considered a main global threat since it occurs in 576 human and animal health” while parts of the article also discussed about data in animal and human. Could the human data also be in the table 1 and 2?
- In term of ESBL E. coli and AMR characterisation, could different methodology affect interpretation of its epidemiology? I would suggest author to also discuss this. Perhaps part of the article could be recommendation on need of harmonisation of methodology and future direction e.g. development and implementation of guideline for AMR etc.
Minor editing of English language required
Author Response
- For the manuscript (including the title), can author confirm that “clonal lineage” is the technically accurate terminology used? as the discussion was mainly only focus on comparison of sequence type but not other e.g. SNP alignment etc. Can author confirm “sequence type” and “clonal lineage” are not different?
Authors’ response:
Clonal lineages were established by PFGE, MLST and agr typing. This was the method we chose to analyze, since MLST (Multilocus sequence type) has gained widespread popularity as a typing method and its use has advanced understanding of bacterial evolution and has provided insights into the epidemiology of bacterial diseases. Being used and studied in most of the studies we reviewed.
- For the sections on prevalence of E. coli/MDR rate, suggest authors to describe the methodology briefly for the review, e.g. sources of articles reviewed, keywords used for the searching, selection/exclusion criteria for articles (e.g. period), analysis approach for the secondary analysis. The description of secondary analysis is important as different ESBL E. coli methodology (e.g. enrichment method/direct culture/PCR or other method) or analysis approach (e.g. definition of MDR i.e. resistant to at least 3 classes of antimicrobials) would need to be taken into consideration when comparison of prevalence/MDR rate.
Authors’ response:
In compliance with this suggestion, table 2 has been improved, presenting now data on ESBL E. coli methodology. Also, a new section, designated as methods has been added, as suggested with the methodology used and more information about these definitions was placed in the review.
- would suggest author to include a key message for each session,. For instance, instead of “antimicrobials use in animal production” can author highlight message eg “antimicrobials use in animal production showing an increasing/tapering trend?; Though varied prevalence of ESBL E. coli between geographical regions, certain ST has been found to be predominating sequence type among the livestock animals ”etc. Quite large portion of the current text includes restating of statistics, it can be quite difficult to follow for key messages of the manuscript. Hence including key messages as a subtitle for each session would definitely help reader to grasp the key messages of the review. For authors’ consideration
Authors’ response:
The text of the whole manuscript has been thoroughly rewritten, to comply with this and with further suggestions.
- Authors in the text mentioned several of background on virulence and pathogenicity of E. coli as justification of this review. E. coli are generally not pathogenic; while concern of AMR is “valid” only when a microorganism causes infections which cannot be treated using the antimicrobials e.g. more severe/invasive salmonellosis requiring treatment. Hence, I believe that a rationale of that E. coli is still studied, would be that it an indicator organism of warm-blooded animals which would allow us to better understand transmission of AMR within livestock or between compartments? (e.g. between different animal species; or between animal and humans; or between animal and environment etc). To this, I would suggest authors to discuss about location of the AMR gene detected (either chromosomally or plasmid-mediated) in E. coli, e.g. to further discuss potential roles of E. coli in “spreading” AMR determinants to other more pathogenic bacteria through horizontal transfer.
Authors’ response:
At the request of other reviewers, the section of background on virulence and pathogenicity of E. coli was removed from the bibliographic review and new and more complete information has been added throughout the review
- Would suggest author to consider the below reference. 2019. Monitoring and surveillance of antimicrobial resistance in bacteria from healthy food animals intended for consumption. Regional Antimicrobial Resistance Monitoring and Surveillance Guidelines –Volume 1. Bangkok
Authors’ response:
We did as suggested and new references inserted.
- The para with the first line 164 needs rephrasing for more clarity. For example, I don’t quite get the below sentences.
- Several studies in Table 1 were carried out all over the world regarding the inclusion of ESBLs-producing E. coli in a diversity of animals as in cattle, swine’s and poultry isolated from fecal samples or food samples more specifically in meat product
- In Guadeloupe, a study made by Gaëlle Gruel et al. in pigs, beef cattle and poul-169 try showed moderate (of what? prevalence?) ESBL-E. coli in pig and beef cattle production, in this island (where?) an anti-170 microbial resistance plan (is author refers to antimicrobial stewardship programme?) on the use of antimicrobials in veterinary has been implemented 171 and the moderate ESBL resistance prevalence probably reflect its effectiveness in the ra-172 tional use of antimicrobials, in broilers, a high frequency of ESBL resistance was observed.
Authors’ response:
The text of the whole manuscript has been thoroughly rewritten, to comply with this and with further suggestions.
- For Table 1. The header of the 6thcolumn needs more clarity, is author refers to “number of samples detected with ESBL-E. coli”. Can author add a column for the table 1 to indicate the method of ESBL e.g. enrichment method, PCR method, Vitek etc, for readers’ consideration when reference for comparison. As it might not be fair comparison without considering the method used.
Authors’ response:
In compliance with this suggestion, table 1 has been improved, presenting now data on methods used for ESBL detected.
- For Table 2. There should be information on isolation year. Suggest author to define MDR, as different papers might refer MDR differently. What means by ‘-‘? Is it refer to that data not available or 0% prevalence—both are different. Authors might want to consider further differentiate the genotype (chromosomally-mediated or plasmid-mediated) as per comment SN4.
Authors’ response:
In compliance with this suggestion, table 2 has been improved, presenting now data on isolation year of the samples. As suggested, the "-" was named as ND, for not determined, since the articles do not have this information. A caption was placed at the end of the table.
- As the conclusion section states “considered a main global threat since it occurs in 576 human and animal health” while parts of the article also discussed about data in animal and human. Could the human data also be in the table 1 and 2?
Authors’ response:
The human part was mentioned in this review since the ST found in samples originating from production animals, are ST that have also been found in some studies that were carried out and used in human samples. It serves only as a comparison and as some strains are already described in a variety of ecological niches, they thus become a public health problem where a health perspective will have to be applied. In the table it does not make much sense to contain this information since I focus exclusively on production animals and this review serves to understand the prevalence and the current situation.
- In term of ESBL E. coli and AMR characterisation, could different methodology affect interpretation of its epidemiology? I would suggest author to also discuss this. Perhaps part of the article could be recommendation on need of harmonisation of methodology and future direction e.g. development and implementation of guideline for AMR etc.
Authors’ response:
The text of the whole manuscript has been thoroughly rewritten, to comply with this and with further suggestions. Especially, the text was revised to improve readability. Table 1 has been improved, presenting now data on methods used for ESBL detected and also, a new section, designated as methods has been added, as suggested with the methodology used and more information about these definitions was placed in the review.
Reviewer 4 Report
This review aims to highlight the clonal lineages and the resistance profiles that are observed in E. coli from food-production animals.
The review gathers relevant and useful information on the subject, however we suggest the inclusion of some information about the phylogenetic groups of E. coli, so that any reader, even those less familiar with the subject, can follow all the information presented, including in table 2.
We also suggest delete prevalence from the title.
Acronyms and abbreviations should be defined the first time they appear in the text: Line 25 – AMR; Line 29 - ESBL-E. coli; Line 98 – UN; Line 299 - MDR; Line 300 – ST; Line 372 – WGS; Line 419 - UTI;
Authors should revise the way some references are cited in the text: Line 169 we recommend “Gruel et al. [29].” and apply the same format with all references cited in this way throughout the manuscript Lines 179, 182, 192, 211, 215, 297, 320, 327, 336, 355, 363, 397, 410, 420, 443, 473, 517 and 528.
In line 292 Authors should refer the exactly significance of MLST - Multilocus Sequence Typing.
In line 294- Include as table 2 caption what MLST, AMR and MDR mean. Tables must be self-interpreting.
In line 328 – Please put E. coli in italics.
In line 346 – Please put sul, sul1 and sul2 and mcr-1 in italics.
In line 453 – Please correct - lincomycin instead of limcomycin
A careful and in-depth review of English by a native speaker is strongly recommended regardless of the suggestions mentioned below:
Line 47-49-48 - “Therefore, serve as a reservoir of antibiotic-resistant bacteria, which can be transferred to human populations through food products, animals or through environment …” Please improve the sentence referring which is the reservoir
Line 54 - “… through various interventions…” instead of “… through various intervention…”
Line 62 – Please check english “Whit the demand for meat..”
Line 65 – “…can increase by up to…” instead of “can be increase up to…”
Line 67 – “…,they are used for metaphylaxis…” Instead of “…,there are used for metaphylaxis…”
Line 68 – “…were diagnosed with disease or showed clinical symptoms…” instead of “..were diagnosed with disease or shown clinical symptoms
Line 69 – “New regulations began in 2022…” instead of “New regulations begin in 2022…”
Line 85 – “ …in livestock feed is considered…” instead of “…in livestock feed has considered…”
Line 86 - “… and in antibiotic residues in foods…” instead of “… and in residual antibiotics in foods…”
Line 93,94 – Please check the sentence “Generally, the antibiotics used as therapeutic are administrated via oral or by infection….”
Line 100 – “The antimicrobials used in pigs include…” instead of “The antimicrobials used in pigs including…”
Line 104 – We suggest “Some studies made in pigs farms demonstrated an abundance of antibiotic resistance genes in pig microbiome and the level of resistance to tetracycline is naturally elevated, in animals raised without the presence of tetracycline.” Instead of “Some studies were made in pigs farms and shown that tetracycline resistance genes were analyzed and demonstrated an abundance of antibiotic resistance genes in pig microbiome and the level of resistance to tetracycline is naturally elevated, in animals raised without the presence of tetracycline”
Line 109 – “ A high prevalence….” Instead of “Also had a high prevalence…”
Line 124 – Please correct “…facultative anaerobic species…” instead of “…facultative anaerobic specie…”
Line 132 – We suggest “The increase on E. coli occurrence has been reported…” instead of “The growth on occurrence of E. coli has been reported…”
Line 134 – “…comparison…” instead of “…comparisons…”
Line 156 – “…confer resistance to third…” instead of “…confer resistance to of third…”
Line 160 – “…extended-spectrum cephalosporin resistance became worrisome..” instead of “…extended-spectrum cephalosporin resistance become worrisome…”
Line 164, 165 – We suggest “…In food-producing animals, the presence of ESBL-producing E. coli has been widespread and there are several studies that have shown the presence of ESBLs in livestock…” instead of “…In food-producing animals, the prevalence of ESBL-producing E. coli has been widespread in the veterinary field and there are several studies that have shown the presence of ESBLs in livestock…”
Line 166 – “…Several studies (Table 1) were carried out…” instead of “…Several studies in Table 1 were carried out…”
Line 166- 169 – Please reformulate the sentence.
Line 173 – Please verify if the resistance in broilers was high or moderate. We suggest that authors use shorter sentences, which help in understanding the text and help to avoid apparent contradictions.
Line 176 - write fecal in lowercase
Line 183 – “…the presence of ESBL was verified in 58.4%...” instead of “…the presence of ESBL is verified in 58.4%...”
Line 197-199 – Please reformulate sentence.
Line 209 – “…such as goat milk…” instead of “…such as got milk…”
Line 212 – “… and chicken meat were studied…” instead of “…and chicken meat was studied..”
Line 214 – “…samples of beef,…” instead of “…samples of meat from beef,…”
Line 228 - “…E. coli is a bacterium that colonizes…” instead of “…E. coli is a bacterium that colonize…”
Line 236 – “…and transfer these resistance genes …” instead of “…and transferred this resistance genes…”
Line 239, 240 – Please reformulate the sentence “At least seven intestinal E. coli pathotypes have been described based on their pathogenetic profile, and are divided into different pathotypes…”
Line 253 – “…encode activities that can be divided…” instead of “…encode activities such as can be divided…”
Line 282 – “…that play various roles…” instead of “…that plays various roles…”
Line 283 – Please check “…invasion, macrophages) cells,…”
Line 291 – “…and present a general summary..” instead of “…and presents a general summary…”
In table 2 – “Feacal” instead of “Feecal” or “Fecal”
Line 301, 302 – Authors refer “they showed resistance for all antimicrobial classes, such as beta-lactamases and fluoroquinolone…”. If they showed resistance to all antimicrobial classes, it is inconsistent to mention only two classes we suggest to delete “such as beta-lactamases and fluoroquinolone”.
Line 308- 310 – Long sentences make it difficult to understand the text. We suggest:
“ 72]. ST744 clones were the most prevalent ST in this study from pigs in UK, and in other studies were detected in seagull samples and wild bird populations in Germany [73]. This suggest wider transmission and recycling and birds may be exposed to isolates from anthropogenic sources due to their scavenging activities[57].” Instead of “72] and ST744 clones were the most prevalent ST in this study from pigs in UK, in other studies detected in other host species in seagull samples and wild bird populations in Germany [73] and suggest wider transmission and recycling, birds may be exposed to isolates from anthropogenic sources due to their scavenging activities [57]…”
Line 323 – “…the resistance frequency was low.” instead “…the frequency were low.”
Line 328, 329 – “…E. coli from food-producing animals were done. A total of 19 pig isolates were studied…” instead of “…E. coli from food-producing animals, a total of 19 pigs isolates were study…”
Line 337, 338 – We suggest “All E. coli isolates from nasal (n = 6) and rectal swabs (n = 5) of healthy pigs processed at abattoirs were ESBL…” instead of “The isolates originating from nasal (n = 6) and rectal swabs (n = 5) from healthy pigs processed at abattoirs and all the E. coli isolates were ESBL….”
Line 340 – “The majority of the isolates harboured…” instead of “The majority of the isolated harboured…”
Line 346 – Please put sul, sul1 and sul2 and mcr-1 in italics.
Line 367 – “…were in agreement with the results of previous studies..” instead of “…were like results of previous studies…”
Line 385 – “The classes of antibiotics…” instead of “The classes of antibiotic..”
Line 386, 387 – “...penicillin and tetracyclines classes were the most commonly used in pigs production..” insread of “…penicillin and tetracyclines class were the most commonly used antibiotic in pigs productions..”
Line 388 – “most common resistances found. When we compare…” instead of “most common resistances found when we compare…”
Line 392 - “…clones more frequent found were…” instead of “…clones more frequent found was…”
Line 396 - “…in this animals were…” instead of “…in this part of the food industry were,,,”
Line 397-399 – Please check this sentence: “In a study carried out by Ama Szmolka et al. in Hungary, 90 E. coli strains obtain from different broiler sources, such as faeces, caecum and bone marrow of broiler chickens and day-old chicks at three different slaughterhouses in North-Central Hungary.” Probably you want to say that 90 E. coli strains were obtained from…
Line 420, 421 – Please improve the sentence “In Italy, a study carried by Elisa Massella et al, in 25 samples obtain from poultry and that were isolated 6 E. coli.”
Line 435 – “…and therefore in a study carried out…” instead “…”and therefore a study carried out…”
Line 459-460 – Please check this sentence and see if this is what you mean.
Line 461—463 – Pease revise the sentence.
Line 466 – “…with clonal strains of E. coli strains…” instead of “…with clonal strains of E. coli…”
Line 473, 474 – “...were 60 faecal samples were collected from diarrhea rabbits on farms, and a total of 55 E. coli strains were isolated.” Instead of “…were 60 faecal samples collected from diarrhea rabbits on farms, in this samples a total of 55 E. coli strains were isolated.”
Line 476 – “…isolates were considered…” instead of “…isolated were considered…”
Line 480 – “…this happens because tetracycline…” instead of “…this happens because the fact that tetracycline…”
Line 486– Please reformulate the sentence “Another study in rabbits was carried out in Portugal in three E. coli isolates with colistin resistance and recovered from intestinal content of necropsied meat rabbits collected from two commercial farms.”
Line 496 – “The identification of colistin resistance was also associated…” instead of “The identification of colistin resistance were also associated…”
Line 506 – “and ST58 were…” instead of “and ST58 was…”
Line 514 – “…the ST131 was also found,…” instead of “…the ST131 has also found,…”
Line 528 – “…where feacal samples from 101 commercial dairy farms were collected.” Instead of “were feacal samples from 101 commercial dairy farms were collected.”
Line 537 – “…could be one of…” instead of “…could have one of ...”
Line 537 - “The clonal lineage,...” instead of “The clonal lineages,…”
Line 559 – “The analysis of the studies in Table 2 demonstrated…” instead of “The analysis of the studies in Table 2, was demonstrate…
Line 562 – Please revise sentence “The most common widespread clones spreading resistance in E. coli are the clonal group A belonging to the D phylogroup and the sequence type ST131 (phylogroup B2), the ST405 (phylogroup D) clones produce extended-spectrum beta-lactamases (ESBLs).
Line 578 - “…cause the emergence and rapid spread of MDR and ESBLs in E. coli…” instead of “…cause the emergence and rapid spread MDR and ESBLs in E. coli…”
Author Response
This review aims to highlight the clonal lineages and the resistance profiles that are observed in E. coli from food-production animals.
- The review gathers relevant and useful information on the subject, however we suggest the inclusion of some information aboutthe phylogenetic groups of coli, so that any reader, even those less familiar with the subject, can follow all the information presented, including in table Authors’response:
The suggested changes were implemented in the revised manuscript.
- We also suggest delete prevalence from the title.
Authors’ response:
We did as suggested.
Acronyms and abbreviations should be defined the first time they appear in the text: Line 25 – AMR; Line 29 - ESBL-E. coli; Line 98 – UN; Line 299 - MDR; Line 300 – ST; Line 372 – WGS; Line 419 - UTI;
Authors’ response:
We did as suggested.
- Authors should revise the way some references are cited in the text: Line 169 we recommend “Gruel et al. [29].” and apply the same format with all references cited in this way throughout the manuscript Lines 179, 182, 192, 211, 215, 297, 320, 327, 336, 355, 363, 397, 410, 420, 443, 473, 517 and 528.
Authors’ response:
We did as suggested.
- In line 292 Authors should refer the exactly significance of MLST - Multilocus Sequence Typing.
Authors’ response:
We did as suggested.
- In line 294- Include as table 2 caption what MLST, AMR and MDR mean. Tables must be self-interpreting.
Authors’ response:
In compliance with this suggestion, a caption in Table 2 was placed at the end of the table.
- In line 328 – Please put coliin italics.
- In line 346 – Please put sul, sul1 andsul2 and mcr-1 in italics.
- In line 453 – Please correct - lincomycin instead of limcomycin
A careful and in-depth review of English by a native speaker is strongly recommended regardless of the suggestions mentioned below:
Authors’ response:
The text of the whole manuscript has been thoroughly rewritten, to comply with this and with further suggestions.
- Line 47-49-48 - “Therefore, serve as a reservoir of antibiotic-resistant bacteria, which can be transferred to human populations through food products, animals or through environment …” Please improve the sentence referring which is the reservoir
- Line 54 - “… through various interventions…” instead of “… through various intervention…”
- Line 62 – Please check english “Whit the demand for meat..”
- Line 65 – “…can increase by up to…” instead of “can be increase up to…”
- Line 67 – “…,they are used for metaphylaxis…” Instead of “…,there are used for metaphylaxis…”
- Line 68 – “…were diagnosed with disease or showed clinical symptoms…” instead of “..were diagnosed with disease or shown clinical symptoms
- Line 69 – “New regulations began in 2022…” instead of “New regulations begin in 2022…”
- Line 85 – “ …in livestock feed is considered…” instead of “…in livestock feed has considered…”
- Line 86 - “… and in antibiotic residues in foods…” instead of “… and in residual antibiotics in foods…”
- Line 93,94 – Please check the sentence “Generally, the antibiotics used as therapeutic are administrated via oral or by infection….”
- Line 100 – “The antimicrobials used in pigs include…” instead of “The antimicrobials used in pigs including…”
- Line 104 – We suggest “Some studies made in pigs farms demonstrated an abundance of antibiotic resistance genes in pig microbiome and the level of resistance to tetracycline is naturally elevated, in animals raised without the presence of tetracycline.” Instead of “Some studies were made in pigs farms and shown that tetracycline resistance genes were analyzed and demonstrated an abundance of antibiotic resistance genes in pig microbiome and the level of resistance to tetracycline is naturally elevated, in animals raised without the presence of tetracycline”
- Line 109 – “ A high prevalence….” Instead of “Also had a high prevalence…”
- Line 124 – Please correct “…facultative anaerobic species…” instead of “…facultative anaerobic specie…”
- Line 132 – We suggest “The increase on colioccurrence has been reported…” instead of “The growth on occurrence of E. coli has been reported…”
- Line 134 – “…comparison…” instead of “…comparisons…”
- Line 156 – “…confer resistance to third…” instead of “…confer resistance to of third…”
- Line 160 – “…extended-spectrum cephalosporin resistance became worrisome..” instead of “…extended-spectrum cephalosporin resistance become worrisome…”
- Line 164, 165 – We suggest “…In food-producing animals, the presence of ESBL-producing colihas been widespread and there are several studies that have shown the presence of ESBLs in livestock…” instead of “…In food-producing animals, the prevalence of ESBL-producing E. coli has been widespread in the veterinary field and there are several studies that have shown the presence of ESBLs in livestock…”
- Line 166 – “…Several studies (Table 1) were carried out…” instead of “…Several studies in Table 1 were carried out…”
- Line 166- 169 – Please reformulate the sentence.
- Line 173 – Please verify if the resistance in broilers was high or moderate. We suggest that authors use shorter sentences, which help in understanding the text and help to avoid apparent contradictions.
- Line 176 - write fecal in lowercase
- Line 183 – “…the presence of ESBL was verified in 58.4%...” instead of “…the presence of ESBL is verified in 58.4%...”
- Line 197-199 – Please reformulate sentence.
- Line 209 – “…such as goat milk…” instead of “…such as got milk…”
- Line 212 – “… and chicken meat were studied…” instead of “…and chicken meat was studied..”
- Line 214 – “…samples of beef,…” instead of “…samples of meat from beef,…”
- Line 228 - “… coliis a bacterium that colonizes…” instead of “…E. coli is a bacterium that colonize…”
- Line 236 – “…and transfer these resistance genes …” instead of “…and transferred this resistance genes…”
- Line 239, 240 – Please reformulate the sentence “At least seven intestinal colipathotypes have been described based on their pathogenetic profile, and are divided into different pathotypes…”
- Line 253 – “…encode activities that can be divided…” instead of “…encode activities such as can be divided…”
- Line 282 – “…that play various roles…” instead of “…that plays various roles…”
- Line 283 – Please check “…invasion, macrophages) cells,…”
- Line 291 – “…and present a general summary..” instead of “…and presents a general summary…”
- In table 2 – “Feacal” instead of “Feecal” or “Fecal”
- Line 301, 302 – Authors refer “they showed resistance for all antimicrobial classes, such as beta-lactamases and fluoroquinolone…”. If they showed resistance to all antimicrobial classes, it is inconsistent to mention only two classes we suggest to delete “suchas beta-lactamases and fluoroquinolone”.
- Line 308- 310 – Long sentences make it difficult to understand the text. We suggest:“ 72]. ST744 clones were the most prevalent ST in this study from pigs in UK, and in other studies were detected in seagull samples and wild bird populations in Germany [73]. This suggest wider transmission and recycling and birds may be exposed to isolates from anthropogenic sources due to their scavenging activities[57].” Instead of “72] and ST744 clones were the most prevalent ST in this study from pigs in UK, in other studies detected in other host species in seagull samples and wild bird populations in Germany [73] and suggest wider transmission and recycling, birds may be exposed to isolates from anthropogenic sources due to their scavenging activities [57]…”
- Line 323 – “…the resistance frequency was low.” instead “…the frequency were low.”
- Line 328, 329 – “… colifrom food-producing animals were done. A total of 19 pig isolates were studied…” instead of “…E. coli from food-producing animals, a total of 19 pigs isolates were study…”
- Line 337, 338 – We suggest “All coli isolates from nasal (n = 6) and rectal swabs (n = 5) of healthy pigs processed at abattoirs were ESBL…” instead of “The isolates originating from nasal (n = 6) and rectal swabs (n = 5) from healthy pigs processed at abattoirs and all the E. coliisolates were ESBL….”
- Line 340 – “The majority of the isolates harboured…” instead of “The majority of the isolated harboured…”
- Line 346 – Please put sul, sul1 andsul2 and mcr-1 in italics.
- Line 367 – “…were in agreement with the results of previous studies..” instead of “…were like results of previous studies…”
- Line 385 – “The classes of antibiotics…” instead of “The classes of antibiotic..”
- Line 386, 387 – “...penicillin and tetracyclines classes were the most commonly used in pigs production..” insread of “…penicillin and tetracyclines class were the most commonly used antibiotic in pigs productions..”
- Line 388 – “most common resistances found. When we compare…” instead of “most common resistances found when we compare…”
- Line 392 - “…clones more frequent found were…” instead of “…clones more frequent found was…”
- Line 396 - “…in this animals were…” instead of “…in this part of the food industry were,,,”
- Line 397-399 – Please check this sentence: “In a study carried out by Ama Szmolka et al. in Hungary, 90 colistrains obtain from different broiler sources, such as faeces, caecum and bone marrow of broiler chickens and day-old chicks at three different slaughterhouses in North-Central Hungary.” Probably you want to say that 90 E. coli strains were obtained from…
- Line 420, 421 – Please improve the sentence “In Italy, a study carried by Elisa Massella et al, in 25 samples obtain from poultry and that were isolated 6 coli.”
- Line 435 – “…and therefore in a study carried out…” instead “…”and therefore a study carried out…”
- Line 459-460 – Please check this sentence and see if this is what you mean.
- Line 461—463 – Pease revise the sentence.
- Line 466 – “…with clonal strains of colistrains…” instead of “…with clonal strains of E. coli…”
- Line 473, 474 – “...were 60 faecal samples were collected from diarrhea rabbits on farms, and a total of 55 colistrains were isolated.” Instead of “…were 60 faecal samples collected from diarrhea rabbits on farms, in this samples a total of 55 E. coli strains were isolated.”
- Line 476 – “…isolates were considered…” instead of “…isolated were considered…”
- Line 480 – “…this happens because tetracycline…” instead of “…this happens because the fact that tetracycline…”
- Line 486– Please reformulate the sentence “Another study in rabbits was carried out in Portugal in three coliisolates with colistin resistance and recovered from intestinal content of necropsied meat rabbits collected from two commercial farms.”
- Line 496 – “The identification of colistin resistance was also associated…” instead of “The identification of colistin resistance were also associated…”
- Line 506 – “and ST58 were…” instead of “and ST58 was…”
- Line 514 – “…the ST131 was also found,…” instead of “…the ST131 has also found,…”
- Line 528 – “…where feacal samples from 101 commercial dairy farms were collected.” Instead of “were feacal samples from 101 commercial dairy farms were collected.”
- Line 537 – “…could be one of…” instead of “…could have one of ...”
- Line 537 - “The clonal lineage,...” instead of “The clonal lineages,…”
- Line 559 – “The analysis of the studies in Table 2 demonstrated…” instead of “The analysis of the studies in Table 2, was demonstrate…
- Line 562 – Please revise sentence “The most common widespread clones spreading resistance in coliare the clonal group A belonging to the D phylogroup and the sequence type ST131 (phylogroup B2), the ST405 (phylogroup D) clones produce extended-spectrum beta-lactamases (ESBLs).
- Line 578 - “…cause the emergence and rapid spread of MDR and ESBLs in coli…” instead of “…cause the emergence and rapid spread MDR and ESBLs in E. coli…”
Authors’response:
The suggested changes were implemented in the revised manuscript.
Reviewer 5 Report
The manuscript is very well written. The language of the manuscript is good despite some small spelling mistakes.
Comments
The title of the manuscript refers to antimicrobials used in food-producing animals. I wonder why fish farming was not included. I suggest authors either include fish farming in the manuscript or adjust the title.
I suggest authors add a paragraph for an overview of clonal lineages.
Others:
Line 230, corrected diseases
line 350, check sentence, missing word
line 463, add "be" after ."..may..."
line 466, " strains of E. coli strains" repetition.
Author Response
Comments
The title of the manuscript refers to antimicrobials used in food-producing animals. I wonder why fish farming was not included. I suggest authors either include fish farming in the manuscript or adjust the title.
Authors’response:
Studies on fish farming were included in this review, namely in the ESBL part. However, I did not find any that made part of the clonal lineages and that the various variants included in table 2 were analyzed.
I suggest authors add a paragraph for an overview of clonal lineages.
Authors’response:
The suggested changes were implemented in the revised manuscript.
Others:
Line 230, corrected diseases
line 350, check sentence, missing word
line 463, add "be" after ."..may..."
line 466, " strains of E. coli strains" repetition.
Reviewer 6 Report
General Comments
The authors’ area of topic is an interesting that they tried to review literatures and summarized information on the AMR prevalence and clonal lineages of E. coli reported from food producing animals. The manuscript is well written, narrated and summarized in tables. Overall, the findings are worthful. However, the manuscript has some important limitations:
1) The authors missed applying and presenting a methodological approach for reviewing existing literatures. I am just curious why the authors did not use and mention the type of review approach followed? Search engines/databases? Search algorithm? Inclusion and exclusion criteria etc. ? This manuscript will be more useful and informative if a systematic review or scoping review method is applied.
2) The authors did not provide an estimate of prevalence of AMR E. coli in different food animals and food of animal origins and the possible variability among animals, foods, different geographic regions etc. The authors summarized previous studies in Table and narrated the individual studies in text in detail which is almost a repetition of the original authors. Even there is no quantitative information on the prevalence of AMR E. coli in the abstract. Honestly speaking it is very difficult to grasp any key take away messages from the review/narration. Had a systematic approach used and meta-analysis applied, then pooled AMR prevalence of E. coli could have been quantified. This would provide more insights on the prevalence, risk factors and variability of AMR E. coli in food producing animals and foods of animal origin at global and regional levels.
3) From the perspective of the authors objective, the heading “Pathogenesis and Virulence factors” is unnecessarily presented.
4) Figure 1 is very shallow and superficial. Given the focus is literature review, the authors should have clearly indicated the potential pathways and presented the conceptual framework for the transmission of AMR pathogens/genes and drivers among the livestock, humans, and environments.
5) Because any systematic approach and extensive review was not used, there are possibilities of missing reporting of some findings elsewhere from all food producing animals. For example, although other considerable number of studies are available, only one study on the prevalence of AMR E. coli from sheep and goats included in the review from a study in Nigeria.
Finally, I strongly suggest the authors should make major revision and address the above points prior to accepting the manuscript for publication.
Author Response
General Comments
The authors’ area of topic is an interesting that they tried to review literatures and summarized information on the AMR prevalence and clonal lineages of E. coli reported from food producing animals. The manuscript is well written, narrated and summarized in tables. Overall, the findings are worthful. However, the manuscript has some important limitations:
1) The authors missed applying and presenting a methodological approach for reviewing existing literatures. I am just curious why the authors did not use and mention the type of review approach followed? Search engines/databases? Search algorithm? Inclusion and exclusion criteria etc. ? This manuscript will be more useful and informative if a systematic review or scoping review method is applied.
Authors’ response:
A new section, designated as methods has been added, as suggested.
2) The authors did not provide an estimate of prevalence of AMR E. coli in different food animals and food of animal origins and the possible variability among animals, foods, different geographic regions etc. The authors summarized previous studies in Table and narrated the individual studies in text in detail which is almost a repetition of the original authors. Even there is no quantitative information on the prevalence of AMR E. coli in the abstract. Honestly speaking it is very difficult to grasp any key take away messages from the review/narration. Had a systematic approach used and meta-analysis applied, then pooled AMR prevalence of E. coli could have been quantified. This would provide more insights on the prevalence, risk factors and variability of AMR E. coli in food producing animals and foods of animal origin at global and regional levels.
Authors’response:
The text of the whole manuscript has been thoroughly rewritten, to comply with this and with further suggestions This section has been completely rewritten, presenting the literature data in a way that leads the reader to the objectives of our literature review. Care has been taken in employing as precise a language as possible.
3) From the perspective of the authors objective, the heading “Pathogenesis and Virulence factors” is unnecessarily presented.
Authors’response:
The suggested changes were implemented in the revised manuscript and heading “Pathogenesis and Virulence factors” has been removed.
4) Figure 1 is very shallow and superficial. Given the focus is literature review, the authors should have clearly indicated the potential pathways and presented the conceptual framework for the transmission of AMR pathogens/genes and drivers among the livestock, humans, and environments.
Authors’ response:
In compliance with this suggestion, new visual material has been added to Figure 1and new information was added to the image to understand it better the transmission of AMR.
5) Because any systematic approach and extensive review was not used, there are possibilities of missing reporting of some findings elsewhere from all food producing animals. For example, although other considerable number of studies are available, only one study on the prevalence of AMR E. coli from sheep and goats included in the review from a study in Nigeria.
Authors’ response:
In compliance with this suggestion, material has been added to table 2 and has been improved, presenting now data on goat and sheep studies.
Round 2
Reviewer 1 Report
Dear Author,
Thanks for editing the paper.
Now, it was improved.
Author Response
We thank the reviewer for the comments and suggestions.
Reviewer 3 Report
Further comments to round 1 review.
- For the sections on prevalence of E. coli/MDR rate, suggest authors to describe the methodology briefly for the review, e.g. sources of articles reviewed, keywords used for the searching, selection/exclusion criteria for articles (e.g. period), analysis approach for the secondary analysis. The description of secondary analysis is important as different ESBL E. coli methodology (e.g. enrichment method/direct culture/PCR or other method) or analysis approach (e.g. definition of MDR i.e. resistant to at least 3 classes of antimicrobials) would need to be taken into consideration when comparison of prevalence/MDR rate.
Authors’ response:
In compliance with this suggestion, table 2 has been improved, presenting now data on ESBL E. coli methodology. Also, a new section, designated as methods has been added, as suggested with the methodology used and more information about these definitions was placed in the review.
- Line 1907 - What is ‘grey literature’?
- Line 1912 – “The tables were created using consistent criteria across all the compiled studies”à can authors list down what exactly are the ‘criteria’ applied?
- keywords used includes “antibiotic resistance”, “livestock animals”, “Esherichia coli”,”antibiotics used in livestock”,”food- 1908 producing animals”, “phylogenetic groups in E. coli”, “ESBL”, and “genetic lineages”. à are these keywords exclusive? For instance, in the text did mention of aquatic animals whereas the keywords only on livestock animals? authors uses ‘antibiotics’ which is a subset of ‘antimicrobials’? will the usage of the keyword would limit the literature to be included in the review? For clarification please.
- Analysis approach for the secondary analysis.--> authors have not included how the analysis was done. Or, is there any secondary analysis?
- As the conclusion section states “considered a main global threat since it occurs in 576 human and animal health” while parts of the article also discussed about data in animal and human. Could the human data also be in the table 1 and 2?
Authors’ response:
The human part was mentioned in this review since the ST found in samples originating from production animals, are ST that have also been found in some studies that were carried out and used in human samples. It serves only as a comparison and as some strains are already described in a variety of ecological niches, they thus become a public health problem where a health perspective will have to be applied. In the table it does not make much sense to contain this information since I focus exclusively on production animals and this review serves to understand the prevalence and the current situation.
- I have difficulty in understanding the scope of the manuscript. The title of the manuscript mentioned ‘One Health’; but now authors explain that the focus of the review is on ‘food-producing animals’.
- If the review is focusing only on production animals, then the title, conclusion should be re-angled. For instance, how can we know occurrence of ESBL E. coli in food producing animals is ‘an emerging One Health concern’ just solely based on occurrence in food-producing animals? The occurrence doesn’t translate directly to ‘public health problem – impact to humans’ if risk assessment (e.g. dose-response, exposure rate) is not done? Also to reiterate, occurrence in food doesn’t mean food is the ‘source’, e.g. is it impossible those STs are from human to animals?
- Can I request to clarify the scope of the manuscript prior to publication? This is to put the content in the right context to avoid misleading readers.
==============================
Just a couple of minor comments further.
- Table 1, Spain, samples is ‘food’. Can authors specify from which food chain points the ‘food’ is collected from? E.g. ‘food (retail/slaughterhouse/farm)’.
- Table 1. to consider format each row in descending order, so that readers can easily know what species is the “top” in that country/region. E.g. Guadeloupe, poultry, followed by beef cattle then pigs.
- Line 408 – the header “2. Food-producing animals as sources of food-borne pathogens: Escherichia coli. 408” – epidemiologically, can we conclude that food-producing animals is the ‘source’ based on occurrence, without studying of the transmission pathway? e.g. animal-animal, or human-animal, or animal-human. Can author re-consider the title?
- Line 980, E. coli should be italicised, please check through the manuscript. Gene names should be italicised, throughout the manuscript. E.g. line 1252 mcr-1 is not italicised
- ‘antibiotics’ and ‘antimicrobials’ are different and should not be used interchangeably. Can author standardise throughout the manuscript?
- after reading through 980 to 1676, I still can’t grasp what’s the predominating STs in overall – they are several lines seems attempting to summarise the predominance but they seem to be restating the prevalence in studies in the table – which readers can see from the table itself. As this is a ‘re-view’, can authors summarise the findings in overall, and reconsider the earlier comments which have not been incorporated, please. [would suggest author to include a key message for each session,. For instance, instead of “antimicrobials use in animal production” can author highlight message eg “antimicrobials use in animal production showing an increasing/tapering trend?; Though varied prevalence of ESBL E. coli between geographical regions, certain ST has been found to be predominating sequence type among the livestock animals ”etc. Quite large portion of the current text includes restating of statistics, it can be quite difficult to follow for key messages of the manuscript. Hence including key messages as a subtitle for each session would definitely help reader to grasp the key messages of the review. For authors’ consideration ]
- They identified several sequence types (ST), 984 with the most prevalent being ST744, ST44, ST88 and ST10. These STs showed resistance 985 for all antimicrobial classes, with the presence of detected antimicrobial resistance (AMR) 986 genes
· The most 1183frequent clones found were ST10, ST744, ST117, ST100, ST88, and ST131
· The sequence types identified in 1282 ESBL-EC isolates varied across the studies, with ST93 being the most common, followed 1283 by ST162, ST48, and ST115 (Figure 2). Clonal
- Table 2, Nigeria row, to correct typo ‘feecal’.
- Some isolates of this clone harbored the blaCTX-M genes, making infections 1041 caused by them difficult to treat in both humans and food-producing animals—what’s the underlying mechanisms that blaCTX-M will make infections difficult to treat? Gene presence doesn’t necessarily correspond to phenotype.
- Plasmid-mediated quinolone resistance (PMQR) genes, in- 1049 cluding qnrS1, aac(6')Ib-cr, and oqxAB, aac(6’)—can confirm if aac(6')Ib-cr is a quinolone gene, not aminoglycoside?
NIL
Author Response
Reviewer 3
- For the sections on prevalence of E. coli/MDR rate, suggest authors to describe the methodology briefly for the review, e.g. sources of articles reviewed, keywords used for the searching, selection/exclusion criteria for articles (e.g. period), analysis approach for the secondary analysis. The description of secondary analysis is important as different ESBL E. coli methodology (e.g. enrichment method/direct culture/PCR or other method) or analysis approach (e.g. definition of MDR i.e. resistant to at least 3 classes of antimicrobials) would need to be taken into consideration when comparison of prevalence/MDR rate.
Authors’ response:
In compliance with this suggestion, table 2 has been improved, presenting now data on ESBL E. coli methodology. Also, a new section, designated as methods has been added, as suggested with the methodology used and more information about these definitions was placed in the review.
- Line 1907 - What is ‘grey literature’?
Authors’ response: The grey literature is produced by entities whose main task is not publishing. Grey literature can include academic papers, including theses and dissertations, research and committee reports, government reports, conference papers, and ongoing research, among others.
- Line 1912 – “The tables were created using consistent criteria across all the compiled studies”à can authors list down what exactly are the ‘criteria’ applied?
Authors’ response: These criteria ensure that the included studies are relevant and meet certain quality standards. In this literature review the criteria applied include relevance to the research question; studies directly related to antibiotic resistance in different livestock animals, E. coli prevalence and genetic lineages in food-producing animals. Publication type and year. The literature search focused on articles published from 2019 to the present. And methodological consistency, if there are variations in methodology among the included studies, these differences are explicitly indicated within the table for a more comprehensive analysis such as: data of the isolation of the samples, total of samples collected, methods used for ESBL detection, Number of samples detected with ESBL-E. coli positive, Prevalence ESBL- E. coli positive (%) (in table 1). In Table 2, different criteria were applied, since the objective was the analysis of the clonal lineages that are disseminated. In this case, the criteria that were applied were also the years of the studies so that they were as recent as possible, the date of isolation of the samples, most prevalent Phylogroup, Clonal lineages found, AMR phenotypes and genotype and MDR (%).
- keywords used includes “antibiotic resistance”, “livestock animals”, “Esherichia coli”,”antibiotics used in livestock”,”food- 1908 producing animals”, “phylogenetic groups in E. coli”, “ESBL”, and “genetic lineages”. à are these keywords exclusive? For instance, in the text did mention of aquatic animals whereas the keywords only on livestock animals? authors uses ‘antibiotics’ which is a subset of ‘antimicrobials’? will the usage of the keyword would limit the literature to be included in the review? For clarification please.
Authors’ response: Aquatic animals were placed in the topic of food-producing animals since they are inserted and designated as farming fish. The use of specific keywords was used to select articles that were within the scope of the literature and can narrow down the focus of the manuscript and ensure that the studies that we include are the more relevant to our topic of interest. What we wanted to cover was only the analysis of resistance to antibiotics in food-producing animals, but articles that were as complete as possible in terms of phenotypic, genotypic characterization and genetic lineages. However, more comprehensive keywords were also included in this review in order not to limit the study and do not become too restrictive and strike a balance between inclusiveness and relevance in the manuscript.
- Analysis approach for the secondary analysis.--> authors have not included how the analysis was done. Or, is there any secondary analysis?
Authors’ response: A comparative analysis and qualitative synthesis was carried out, which was synthesized in the tables that were created. Where the aim was to identify common patterns and concepts across different studies to develop a deeper understanding of the research topic and also to compare and contrast findings from different studies in order to identify similarities and differences between studies carried out in different parts of the world. We did as suggest and this information have been added.
- As the conclusion section states “considered a main global threat since it occurs in 576 human and animal health” while parts of the article also discussed about data in animal and human. Could the human data also be in the table 1 and 2?
Authors’ response:
The human part was mentioned in this review since the ST found in samples originating from production animals, are ST that have also been found in some studies that were carried out and used in human samples. It serves only as a comparison and as some strains are already described in a variety of ecological niches, they thus become a public health problem where a health perspective will have to be applied. In the table it does not make much sense to contain this information since I focus exclusively on production animals and this review serves to understand the prevalence and the current situation.
- I have difficulty in understanding the scope of the manuscript. The title of the manuscript mentioned ‘One Health’; but now authors explain that the focus of the review is on ‘food-producing animals’.
- If the review is focusing only on production animals, then the title, conclusion should be re-angled. For instance, how can we know occurrence of ESBL E. coli in food producing animals is ‘an emerging One Health concern’ just solely based on occurrence in food-producing animals? The occurrence doesn’t translate directly to ‘public health problem – impact to humans’ if risk assessment (e.g. dose-response, exposure rate) is not done? Also to reiterate, occurrence in food doesn’t mean food is the ‘source’, e.g. is it impossible those STs are from human to animals?
- Can I request to clarify the scope of the manuscript prior to publication? This is to put the content in the right context to avoid misleading readers.
Authors’ response: The title was reformulated, and the one health part was removed, thus maintaining the focus on production animals which is effectively the objective of this manuscript. In the conclusion the part relating to "an emerging One Health concern" was also removed as suggested. Effectively occurrence in food doesn't mean food is the 'source', and the source could be any of the environments since everything is interconnected. We just wanted to show that the clonal lineages found are no longer exclusive to production animals or humans and that some of them were reported for the first time in production animals and that nowadays they are also found to be disseminated in samples that come from humans. However, as you requested, this part was removed so that the purpose of this manuscript is clearer to the reader.
Just a couple of minor comments further.
- Table 1, Spain, samples is ‘food’. Can authors specify from which food chain points the ‘food’ is collected from? E.g. ‘food (retail/slaughterhouse/farm)’.
Authors’ response: We did as suggested.
- Table 1. to consider format each row in descending order, so that readers can easily know what species is the “top” in that country/region. E.g. Guadeloupe, poultry, followed by beef cattle then pigs.
Authors’ response: We did as suggested.
- Line 408 – the header “2. Food-producing animals as sources of food-borne pathogens: Escherichia coli. 408” – epidemiologically, can we conclude that food-producing animals is the ‘source’ based on occurrence, without studying of the transmission pathway? e.g. animal-animal, or human-animal, or animal-human. Can author re-consider the title?
Authors’ response: The title has been re-consider as suggested
- Line 980, E. coli should be italicised, please check through the manuscript. Gene names should be italicised, throughout the manuscript. E.g. line 1252 mcr-1 is not italicized
Authors’ response: The manuscript has been checked throughout and italics have been added with was missing
- ‘antibiotics’ and ‘antimicrobials’ are different and should not be used interchangeably. Can author standardise throughout the manuscript?
Authors’ response :We did as suggested.
- after reading through 980 to 1676, I still can’t grasp what’s the predominating STs in overall – they are several lines seems attempting to summarise the predominance but they seem to be restating the prevalence in studies in the table – which readers can see from the table itself. As this is a ‘re-view’, can authors summarise the findings in overall, and reconsider the earlier comments which have not been incorporated, please. [would suggest author to include a key message for each session,. For instance, instead of “antimicrobials use in animal production” can author highlight message eg “antimicrobials use in animal production showing an increasing/tapering trend?; Though varied prevalence of ESBL E. coli between geographical regions, certain ST has been found to be predominating sequence type among the livestock animals ”etc. Quite large portion of the current text includes restating of statistics, it can be quite difficult to follow for key messages of the manuscript. Hence including key messages as a subtitle for each session would definitely help reader to grasp the key messages of the review. For authors’ consideration ]
Authors’ response: In the table are all the ST that were detected in the study, in the text the most predominant ST in each study is summarized and it concludes with a comparison of all the studies, for each production animal and this is how the table was divided by animal where several studies in different regions are analyzed in order to be able to verify similarities in the results of the different studies. Key message for each session were placed as subtitle throughout the text as suggested.
- They identified several sequence types (ST), 984 with the most prevalent being ST744, ST44, ST88 and ST10. These STs showed resistance 985 for all antimicrobial classes, with the presence of detected antimicrobial resistance (AMR) 986 genes
Authors’ response :In the table are all the ST that were identified in the study (ST44, ST88, ST10, ST744, ST58, ST117, ST48, ST2721) in question, in the text I identified which were the most prevalent ST and which were detected in a larger number of isolates that were ST744, ST44, ST88 and ST10.
- The most 1183frequent clones found were ST10, ST744, ST117, ST100, ST88, and ST131
- The sequence types identified in 1282 ESBL-EC isolates varied across the studies, with ST93 being the most common, followed 1283 by ST162, ST48, and ST115 (Figure 2). Clonal
Authors’ response: Regarding the most frequent ST, in the analysis of the various studies we found that these were the most frequently detected in several isolates. In each study analyzed, it was possible to observe which ST were found and also a detailed analysis of each isolate. That's how we got this information. The information that is in the table relative to the ST, is the information of all the ST that were found in the same study and not the most frequent ones.
- Table 2, Nigeria row, to correct typo ‘feecal’.
Authors’ response: We did as suggested.
- Some isolates of this clone harbored the blaCTX-M genes, making infections 1041 caused by them difficult to treat in both humans and food-producing animals—what’s the underlying mechanisms that blaCTX-M will make infections difficult to treat? Gene presence doesn’t necessarily correspond to phenotype.
Authors’ response: The presence of blaCTX-M genes determines that we are dealing with Extended-spectrum beta-lactamase (ESBL)-producing bacteria that allows it to become resistant to a wide variety of penicillins and cephalosporins. It is also described that CTX-M-encoding genes have been captured from the chromosome of Kluyvera spp. on conjugative plasmids that mediate their dissemination among pathogenic enterobacteria and also the production of ESBLs is one of the primary mechanisms conferring resistance to broad-spectrum β-lactam antibiotics and therefore the antibiotics available to use in the treatment are more limited and in this way will make infections difficult to treat.
- Plasmid-mediated quinolone resistance (PMQR) genes, in- 1049 cluding qnrS1, aac(6')Ib-cr, and oqxAB, aac(6’)—can confirm if aac(6')Ib-cr is a quinolone gene, not aminoglycoside?
Authors’ response: Several articles describe the aac(6′)-Ib-cr gene as being the most prevalent plasmid-mediated fluoroquinolone (FQ) resistance mechanism in Enterobacteriaceae. It is also that the PMQR genes are associated with resistance to quinolones, cephalosporins and aminoglycosides.
Reviewer 6 Report
The authors significantly improved the manuscript.
Author Response

(The authors gave the same response as above.)
